# Pre-deliquescent water uptake in deposited nanoparticles observed with *in situ* ambient pressure X-ray photoelectron spectroscopy

Jack J. Lin[1], Kamal Raj R[1], Stella Wang[2], Esko Kokkonen[3], Mikko-Heikki Mikkelä[3], Samuli Urpelainen[1], and Nønne L. Prisle[1]

[1]Nano and Molecular Systems Research Unit, P. O. Box 3000, FI-90014 University of Oulu, Finland
[2]Division of Physics, Math, and Astronomy, California Institute of Technology, Pasadena, California, 91125, USA
[3]MAX IV Laboratory, Lund University, Box 118, SE-22100 Lund, Sweden

**Correspondence:** Nønne L. Prisle (nonne.prisle@oulu.fi); Samuli Urpelainen (samuli.urpelainen@oulu.fi)

**Abstract.** We study the adsorption of water onto deposited inorganic sodium chloride and organic malonic acid and sucrose nanoparticles at ambient water pressures corresponding to relative humidities (RH) from 0 to 16 %. To obtain information about water adsorption at conditions which are not accessible with typical aerosol instrumentation, we use surface-sensitive ambient pressure X-ray photoelectron spectroscopy (APXPS), which has a detection sensitivity from parts per thousand. Our results show that water is already adsorbed on sodium chloride particles at RH well below deliquescence, and that the chemical environment on the particle surface is changing with increasing humidity. While the sucrose particles exhibit only very modest changes on the surface at these relative humidities, the chemical composition and environment of malonic acid particle surfaces is clearly affected. Our observations indicate that water uptake by inorganic and organic aerosol particles could already have an impact on atmospheric chemistry at low relative humidities. We also establish the APXPS technique as a viable tool for studying chemical changes on the surfaces of atmospherically relevant aerosol particles which are not detected with typical online mass- and volume-based methods.

*Copyright statement.* TEXT

## 1 Introduction

The interaction between atmospheric particulate matter and water is one of the most important processes in Earth's atmosphere. The amount of water associated with an aerosol particle is a function of its intrinsic hygroscopicity and the ambient relative humidity (RH). Liquid water comprises a significant fraction of the global aerosol mass with field measurements documenting the global presence of the metastable aerosol phase state (Rood et al., 1989; Nguyen et al., 2016). Condensed water in the atmosphere critically influences both direct and indirect climate effects of aerosols, governed by aerosol growth and light scattering and via the activation of aerosol particles into cloud droplets (Kreidenweis and Asa-Awuku, 2014).

The surfaces of aerosol particles and droplets are distinct physical and chemical environments compared to their associated bulk phases. Reaction rates in micrometer-scale droplets have been measured (Jacobs et al., 2017; Marsh et al., 2019; Zhang

et al., 2020b) and modeled (Benjamin, 2019; Mallick and Kumar, 2020) to be higher than those in bulk water with some reactions even proceeding spontaneously (Lee et al., 2019). For cloud and fog systems where the interfacial region makes up a significant fraction of the condensed aqueous phase, the reaction rate at the surface can be the rate-limiting step in multi-phase

OH oxidation involving surface-active organic species such as pinonic acid (Huang et al., 2018). Interfacial water molecules can promote reactions between organic acids and $SO_3$, which are distinct from those that occur in the gas phase and important for heterogeneous formation of $H_2SO_4$ and subsequent new particle formation (Zhong et al., 2019; Lv and Sun, 2020). The bulk/surface partitioning of saccharide molecules in aqueous solution affects their ability to be oxidized by OH (Fan et al., 2019). The presence of surface-enriched nonanoic acid also increases the surface concentration of Fe(III) in aqueous mixtures

which mediates enhanced photochemical release of volatile organic compounds (Huang et al., 2020).

The composition of the droplet surface can influence the mass transport and chemical reactions that occur at the surface (e.g. Cosman et al., 2008; Park et al., 2009; Roy et al., 2020). The acidity of organic acids on water surfaces has been measured to be much lower than predicted for the bulk aqueous phase (Enami et al., 2010; Prisle et al., 2012; Öhrwall et al., 2015; Werner et al., 2018). Uptake of glyoxal has been shown to be enhanced at acidic interfaces (Shi et al., 2020). Changes in

droplet size via evaporation or condensation can affect the chemical reactions that take place at the surface via changes in the relative dimensions and chemical environments of the interface and associated bulk phases (Prisle et al., 2012; Djikaev and Ruckenstein, 2019; Petters et al., 2020). The presence of surface-active organic molecules on droplet surfaces can also affect droplet surface tension (Shulman et al., 1996; Prisle et al., 2010b; Bzdek et al., 2020) and morphology (Kwamena et al., 2010) that affect both warm (Sareen et al., 2013; Ovadnevaite et al., 2017) and ice cloud nucleation (Knopf and Forrester, 2011;

Perkins et al., 2020) as well as droplet coalescence (Pak et al., 2020).

The formation of an aqueous aerosol phase can lead to the partitioning of water-soluble gases to the condensed phase (Prisle et al., 2010a), including many reactive oxidants (Donaldson and Valsaraj, 2010), that can initiate a wide range of aqueous phase chemistry (McNeill, 2015). The mobilization of ions in aqueous solution has the ability to influence atmospheric chemistry (Cwiertny et al., 2008). In particular, the mobilization of chloride ion from sea salt aerosol is a significant source of chlorine gas

in the troposphere, with the subsequent formation of chlorine radicals affecting the budgets of important atmospheric species such as volatile organic compounds, ozone, OH, and nitrogen oxides (Wang et al., 2019). Recent field studies have detected the depletion of chloride and bromide from marine aerosol particles under the influence of acidic species from wildfire emissions (Braun et al., 2017). A number of aqueous phase reactions occur between inorganic salt species and organic compounds. The hygroscopic properties of sodium halide particles coated with fatty acids depend on both the salt anion and the carboxylic acid,

with some mixtures showing barriers to water uptake while others do not (Miñambres et al., 2014). Enhanced production of sulfate aerosol via nitrate photolysis was observed to be facilitated by the presence of surface-active halide ions (Zhang et al., 2020a).

A wide range of experimental techniques are available to study the hygroscopic properties of particles in sub-saturated (< 100 % RH) conditions, including humidity-tandem differential mobility analyzers (H-TDMA), various optical extinction and

scattering methods, physisorption analyzers, quartz crystal microbalances and various microscopic and spectroscopic methods (Kreidenweis and Asa-Awuku, 2014; Tang et al., 2019). A drawback of many of the otherwise highly valuable traditional

aerosol techniques, such as the H-TDMA or optical methods, is that they are based on detection of growth in mass or volume. When applied alone, these methods are not sensitive enough for detecting minute amounts of water adsorbed onto the aerosols at low humidity or below the deliquescence relative humidity (DRH). Furthermore, most of these methods do not provide

molecular-level information about the system and are insensitive to surface-specific chemistry and the phase state of adsorbed water. This can lead to underestimating the amount and impact of water adsorption at lower RH. The equilibrium between water and the particle phase has been shown experimentally to depend on the particle phase state and humidity history, with important implications for particle hygroscopic growth and cloud droplet activation (Bilde and Svenningsson, 2004).

A number of spectroscopic techniques have been applied to systems of environmental and atmospheric relevance (see e.g. the

reviews of Ault and Axson, 2016; Tang et al., 2019). Surface-sensitive techniques such as X-ray photoelectron spectroscopy (XPS) provide molecular-level information on the chemical composition and properties of surfaces. So far, most of these studies, in particular those employing XPS, have adopted a surface science approach using single crystal surfaces as simplified model systems (Tang et al., 2019). For example, XPS has been used to study the photochemistry of $TiO_2$ (110) (Lampimäki et al., 2015), ionic mobility on NaCl (001) surfaces induced by water adsorption (Verdaguer et al., 2008) and reaction of water

vapor with MgO (001) surfaces (Kaya et al., 2011). XPS on a liquid micro-jet (Winter, 2009) has also been used to study surface-specific chemistry of aqueous solutions comprising atmospherically relevant organic surfactants (Prisle et al., 2012; Werner et al., 2014; Öhrwall et al., 2015; Walz et al., 2015, 2016; Toribio et al., 2018; Werner et al., 2018; Ammann et al., 2018) as macroscopic model systems for atmospheric microscopic water droplets. Experiments on single aerosol particles are, on the other hand, scarce (Antonsson et al., 2015; Shakya et al., 2016; Ouf et al., 2016). As some of the few examples of atmospheric

relevance, XPS has has been employed to study the surface composition of size-segregated ambient aerosol collected from an urban environment (Cheng et al., 2013). We have previously studied solvation of RbBr in free flying water clusters (Hautala et al., 2017b), as well as size-dependent structural phase changes in CsBr (Hautala et al., 2017a), as model systems for salt clusters of higher environmental relevance. Surface-enhanced Raman spectroscopy (SERS) has also been employed to study the surface composition of atmospherically relevant particles generated via electrospray (Gen and Chan, 2017; Gen et al.,

2019). Using environmental transmission electron microscopy, Wise et al. (2008) observed significant amounts of water uptake onto NaCl particles at humidities as low as 70 % RH, well below the deliquescence RH. With the advent of high-brilliance synchrotron radiation and recent developments in electron analyzer technology, previous barriers of too low sample density can be overcome, enabling XPS studies under realistic conditions, *in situ* and *operando*, using the so-called ambient pressure XPS (APXPS) technique (Salmeron and Schlögl, 2008; Ogletree et al., 2009; Starr et al., 2013; Kong et al., 2020).

In this work, we study the pre-deliquescent water adsorption to laboratory-generated, deposited sodium chloride (NaCl), malonic acid, and sucrose particles. NaCl is a major component of sea salt, which is the most abundant aerosol species by mass in the atmosphere (Murphy et al., 2019). Salt aerosols, apart from being important light scatterers, also take part in atmospheric chemistry by interacting with atmospheric trace gases as a source of halogens (Rossi, 2003), especially in the presence of water in the aerosols. The water uptake and deliquescence of salt aerosol particles can be affected by the presence of other

inorganic or organic matter. For example, the presence of malonic acid in NaCl aerosol will not only lower the DRH of the aerosol, but also facilitate the depletion of chlorine (Laskin et al., 2012). Organic material comprises a large fraction of the

ambient aerosol mass (Kanakidou et al., 2005), and malonic acid and sucrose represent hygroscopic organic compounds with different chemical functionalities. Dicarboxylic acids such as malonic acid have been identified in ambient aerosols samples where they can dominate the water-soluble organic fraction (Khwaja, 1995; Yu et al., 2005; Decesari et al., 2000, 2001) and are

hygroscopic in both sub- and supersaturated conditions (Prenni et al., 2001; Hori et al., 2003; Rissman et al., 2007; Pope et al., 2010). Sucrose is studied as a model carbohydrate that can form a glassy state in response to changing relative humidity with effects on its ability to act as either cloud or ice nuclei (Zobrist et al., 2011; Estillore et al., 2017). Utilizing APXPS, interaction with water vapor at the particle surface was studied at relative humidities between 0 and 16 %. To the best of our knowledge, this is the first time XPS has been used on sampled nano-scale particles of immediate atmospherically relevant composition at

ambient pressure conditions.

## 2 Experimental

We employed the APXPS technique to obtain chemically specific information about the composition of the nanoparticle surfaces. Photoelectron spectroscopy utilizes the photoelectric effect, by which the sample is ionized from inelastic collisions with photons and the emitted electrons are detected and characterized in terms of their kinetic energy ($E_k$). When the ionizing pho-

ton energy ($h\nu$) is known, the binding energy ($E_b$) of electrons within the sample can be determined simply as $E_b = h\nu - E_k$. By using X-ray photons, core-level atomic-like orbitals can be ionized, and the electron binding energy gives a very sensitive fingerprint of the chemical composition and environment of the sampled region. XPS is furthermore a highly surface-sensitive technique, because the resulting kinetic energies of the photoelectrons yield very short characteristic attenuation lengths and the detected photoelectron signal therefore originates mainly from the topmost few nanometers of the sample. An XPS measure-

ment consists of measuring the intensity of photoelectrons emitted from the sample as a function of the characterized electron kinetic energy. Typically, an XPS spectrum presents the photoelectron signal intensity as function of the orbital binding energy and consists of a collection of peaks, each corresponding to a different chemical species or specific environment, which is identified by the spectral position in terms of the binding energy. Here, we quantify the spectral peaks in terms of their areas, which are directly proportional to the relative abundances of each species or environment on the surface of the sample. Spectral

fitting techniques are employed to obtain accurate results for both binding energies and peak areas.

Experiments were carried out at the APXPS end station (Schnadt et al., 2012; Knudsen et al., 2016) of the SPECIES beamline (Urpelainen et al., 2017) at the MAX IV Laboratory in Lund, Sweden. The end station is equipped with a hemispherical SPECS Phoibos NAP-150 electron energy analyzer and allows for measurements both at ultra-high vacuum (UHV) conditions, as well as up to 25 mbar in an ambient pressure (AP) cell. During the experiments, the SPECIES beamline was still under construction,

and we used a double anode (SPECS XR-50) X-ray source with Al (1486.6 eV) and Mg (1253.6 eV) anodes for exciting the samples instead of synchrotron radiation.

Samples were prepared at the Lund University Aerosol Lab and deposited on either silicon or gold substrates. Deposited samples were kept and transferred to the SPECIES end station in a desiccator in order not to expose them to ambient humidity for extended periods of time prior to the experiments. For the XPS measurements, the substrates were mounted onto stainless

steel sample holders using an adhesive Cu tape in a clean tent environment and loaded into the end station. The experiments consisted of recording C 1s, O 1s (sucrose and malonic acid), Na 1s and Cl 2p (NaCl) core level XPS spectra of the deposited aerosol particles at UHV and *in situ* at varying RH conditions. The Al anode was used to measure spectra from NaCl while the Mg anode was used to measure spectra from the malonic acid and sucrose samples.

A detailed account of the sample preparation process and experimental conditions is given in the following sections.

## 2.1 Sample preparation

Aerosol samples were generated by nebulizing aqueous solutions of either sodium chloride (NaCl), sucrose ($C_{12}H_{22}O_{11}$), or malonic acid ($CH_2(COOH)_2$). Solutions were prepared using ultra-pure Milli-Q water. All chemicals were obtained from Sigma-Aldrich and used without further treatment.

The air flow containing solution droplets (3 L min$^{-1}$) was mixed with dry, particle-free air (3 L min$^{-1}$) in a 3 L aerosol mixing chamber. From the mixing chamber, 1–1.5 L min$^{-1}$ was sent through a diffusion dryer followed by a $^{63}$Ni bipolar charger. The dried and charge neutralized aerosol flow was divided between a scanning mobility particle sizer (SMPS; TSI 3936) for size distribution characterization and a nanometer aerosol sampler (NAS; TSI 3089) for sample collection of the entire dry aerosol particle size distribution. Sodium chloride particles were collected onto silicon (with native oxide) wafers, while sucrose and malonic acid particles were deposited onto a gold film substrate.

The surface coverage on the substrate in the NAS is a function of the particle concentration in the gas flow, the drift velocity of the particles in the electric field, and the total sampling time (Preger et al., 2020). Particles are assumed to be spherical across the entire size distribution with a charge distribution described by Boltzmann statistics. We further assume uniform, 100 % deposition efficiency of the positively charged fraction with values above 100 % indicating the presence of more than one monolayer of deposited particles. Details of the sample generation are given in Table 1.

**Table 1.** Aerosol sampling data including generated size distribution information (geometric mean number $\mu_N$, geometric number standard deviation $\sigma_g$, total number $N$, and geometric mean surface area $\mu_{SA}$) and sample collection parameters (sampler flow rate $Q$, collection time $t$, substrate, and coverage).

| Compound | Mean Size Distribution | | | | Sample Collection | | | |
|---|---|---|---|---|---|---|---|---|
| | $\mu_N$ (nm) | $\sigma_g$ | N (cm$^{-1}$) | $\mu_{SA}$ (nm$^2$) | Q (L min$^{-1}$) | t (min) | Substrate | Coverage (%) |
| Sodium chloride | 72 | 2.0 | $3.94 \times 10^5$ | 195 | 1.1 | 42 | Si | 232 |
| Sucrose | 79 | 2.2 | $3.59 \times 10^4$ | 202 | 1.1 | 90 | Au | 56 |
| Malonic acid | 52 | 2.1 | $1.02 \times 10^5$ | 193 | 1.2 | 275 | Au | 245 |

## 2.2  XPS measurements

UHV XPS spectra were recorded using a pass energy of 50 eV and an entrance slit of 3 mm $\times$ 20 mm. This contributes to an experimental broadening of approximately 500 meV in addition to the natural broadening from the excitation source (850 meV for Al K$_\alpha$ and 680 meV for Mg K$_\alpha$ radiation).

For measurements at humid conditions, samples were transferred from the UHV manipulator to the AP cell. Milli-Q water vapor was let into the AP cell through a high precision leak valve. The pressure inside the cell was kept constant using a valve connected to a pump in a feedback loop with an absolute capacitance manometer. All measurements were made at 25 °C at which the saturation vapor pressure of water is $P_{sat}$ = 31.73 mbar. The relative humidity was calculated using the (water) vapor pressure inside the AP cell so that RH = $P_{cell}/P_{sat}$. While samples were inside the cell, spectra were recorded at different water vapor pressure conditions (NaCl: 0, 2, 5, and again at 0 mbar; sucrose 0.2, 1, and 5 mbar; malonic acid 0.2 and 1 mbar). The water vapor pressures of 0, 0.2, 1, 2, and 5 mbar correspond to relative humidities of 0, 0.63, 3.2, 6.3, and 16 %, respectively. These relative humidities are well below the DRH for NaCl (75.3 % (Tang and Munkelwitz, 1993)), malonic acid (72.1 % (Parsons et al., 2004)), and sucrose (85.7 % (Yao et al., 2011)). Any remaining air was removed from the water by several freeze-pump-thaw cycles before introducing the vapor into the AP cell. The cell was also purged with dry nitrogen gas in order to remove any excess water after the experiment. In the case of NaCl particles, the sample was heated up to 125 °C using a button heater placed behind the sample holder. This was done after the sample was dosed up to 10 mbar to see if the changes incurred on exposure to water vapor were reversible. Spectra obtained at 10 mbar, however, are not included in the analysis below due to very low signal to noise ratios.

The electron count rates inside the AP cell are reduced when compared to measurements in UHV conditions due to the attenuation of X-ray intensity by the SiN$_3$/Al window, the limited aperture of the differential pumping of the electron analyzer and scattering of the photoelectrons from the vapor (Knudsen et al., 2016). In order to increase the count rates, spectra recorded inside the cell were acquired at a pass energy of 100 eV instead of 50 eV as used in UHV conditions in order to compensate for the reduced intensity. The analyzer broadening when using 100 eV pass energy was approximately 1.00 eV. Each spectrum was recorded 50-100 times in static conditions for increased statistics. To check the reproducibility of the measurements, we first verified that the individual spectra had not drifted in energy with time, after which all spectra recorded for a given sample and humidity condition were averaged for the final representation.

## 2.3  Data Analysis

Recorded XPS spectra were fitted in order to accurately determine the binding energies of the core electrons and the relative amounts of the probed elements under different conditions. Chemical identities and environments were identified from binding energies obtained from the XPS peak positions and their relative amounts by determining the corresponding peak areas. Data analysis was performed using the Igor Pro software (WaveMetrics, Inc, Lake Oswego, OR, USA). A Shirley-type background was removed from the data before fitting the peaks using the SPANCF curve fitting macro package (Kukk et al., 2001, 2005). All spectra were fitted using symmetric Voigt line shapes. A linear background was included in all the fits to remove any

residual background after the Shirley-type background removal. The energy scales of the spectra were calibrated using the well-known binding energies of Si 2p and Au 4f recorded for the silicon and gold substrates, respectively, as the low aerosol

particle coverage allowed simultaneous measurement of the substrate.

The aim of the spectral fitting procedure is to characterize the measured XPS spectra in terms of peak position and area. The position of a given peak gives the binding energy of the core electron for each probed element. Changes in the binding energy as well as the width of the fitted peaks–or peak broadening–can indicate changes in the chemical environment or physical state of the sampled surface. The area of the peak is directly proportional to the amount of the element being measured. For the

analysis here, we determine the elemental composition of particle surfaces as the relative ratios of the core level peak areas. The peak area of the XPS signal depends on a number of factors, including experimental parameters of the incident radiation and electron spectrometer as well as physical and environmental properties affecting the orbital from which the photoelectron originated. If all of these parameters are known, the XPS signal can be used to quantify the amount of species $i$. While these parameters are not always known, comparison of XPS signals is still possible to quantify relative differences in elemental

abundances and chemical states between experimental conditions. Before extracting relative ratios of the peaks, all spectra were normalized to the photoionization cross section (Yeh and Lindau, 1985) of the given core electron. The attenuation of photoelectron intensity due to scattering of the photoelectrons from the water vapor was estimated by using the kinetic theory formulation (Ogletree et al., 2009) and measured electron scattering cross section data (Muñoz et al., 2007). The attenuation must be taken into account, because the fixed excitation energy from the X-ray source leads to significantly different kinetic

energies of the emitted photoelectrons and consequently different mean free paths in the vapor environment.

## 3    Results and discussion

Below we present experimental XPS spectra together with details and results of curve fitting for deposited NaCl, sucrose and malonic acid aerosol nanoparticles. Binding energy shifts of varying degrees from the dry conditions are observed with increasing relative humidity, indicating changes in the chemical environment of the target element. In addition, changes in the

stoichiometry of molecules on the particle surface can be seen from the intensity ratios between different core level XPS peaks. This is a sign of changes in the chemical composition on the surface of the deposited particles. These observations are clear indications that water is already being adsorbed onto the particles and interacting with molecules on their surfaces at these low relative humidity conditions. The implications and possible interpretations of these observations are discussed below.

### 3.1   NaCl

Figures 1 and 2 show the recorded Na 1s and Cl 2p spectra of NaCl aerosol particles deposited on a Si substrate measured at 0, 2, 5, and again at 0 mbar (0, 6.3, 16, and 0 % RH, respectively) water vapor pressure. Spectra recorded at UHV are shown in Figs. S2 and S3 of the supplement, since they provide the same information as the measurements made in the AP chamber at 0 mbar water vapor pressure. The Na 1s spectra were fitted using a single, symmetric Voigt peak, except for the 0 mbar spectrum after water exposure, where two peaks were required in order to obtain a fit that represents the measured spectrum. The Cl 2p

spectra were fitted with two peaks representing the 1/2 and 3/2 spin-orbit components. The O 1s region for the NaCl sample was also monitored during the experiments, but the spectra are dominated by the signals from the native oxide of the Si wafer and the vapor phase water, and no direct signs of adsorbed or liquid water could therefore be observed from these spectra at low RH. These spectra are therefore omitted from the discussion below.

After calibrating the energy scale using the substrate Si 2p peaks, the binding energy of the Na 1s core level at UHV
conditions was determined to be 1073.48 eV and is indicated by the dashed vertical line in Fig. 1. The signal-to-noise ratio in the spectra recorded at 5 mbar (16 % RH) is very low and the fitted line should be regarded more as guiding the eye than an accurate fit to the peak. The binding energy of the Cl 2p 3/2 component at UHV conditions was determined to be 200.3 eV shown by the dashed vertical line in Fig. 2.

The binding energies measured here for Na 1s and Cl 2p in deposited NaCl nanoparticles at UHV conditions are higher than
the values typically reported in the literature (e.g. Beard, 1993). When binding energy shifts are determined for humidified conditions, they are calculated relative to the binding energy measured here at 0 mbar, and not relative to the literature value. The loss of electrons due to photoionization from the X-ray beam leads to positive charging of the sample surface since the silicon substrate is a semiconductor. The charging of the sample causes an apparent increase in the binding energies seen here. The addition of ambient gas during APXPS measurements can offset the effect of surface charging since photoelectrons from
the photoionization of the gas can travel to the sample surface and compensate for the positive charge.

### 3.1.1 Binding energy shifts: changes in chemical environment

Figure 3 shows the binding energy shifts relative to dry conditions of the Na 1s and Cl 2p core level peaks (red diamonds and dots, respectively) as a function of water vapor pressure at 25 °C. We observe a significant shift of approximately 1.1 eV in both Na 1s and Cl 2p binding energies already when water vapor is introduced into the system at very low relative humidities.
The observed shifts are greater than the error estimates in the peak positions shown in Fig. 3 and therefore cannot be entirely explained by measurement or fitting uncertainty. After this first rapid decrease, the Cl 2p binding energies reach a plateau, when the relative humidity is increased to 16 %. The binding energy shifts for Na 1s and Cl 2p between 0 % and 6.3 % RH are nearly identical, but the subsequent shift for Na 1s between 6.3 % and 16 % RH is towards larger binding energies by 0.17 eV, while Cl 2p shifts towards lower binding energies by -0.26 eV.
Some of the shift in binding energy can be attributed to the introduction of water vapor lessening the effect of charging of the sample compared to the measurements at 0 mbar, but since the amount of ambient gas changes between the measurements, we cannot deconvolute the individual effects of condensed water and gas-phase water on the binding energy shift. However, the binding energy shifts observed here are in good agreement with those reported for Na 2s and Cl 2p in NaCl (001) single crystal surfaces by Verdaguer et al. (2008), even showing a similar plateau at intermediate pressures. The measurements by
Verdaguer et al. (2008) were done at constant water vapor pressure and the similarities in the observed binding energy shifts with relative humidity between their work and ours therefore suggest that condensed water also plays a role in the binding energy shift observed in the present measurements. As Verdaguer et al. (2008) did not report the magnitude of the binding energy shift from 0 to 5 % RH, we have shifted their 5 % RH data point to coincide with our data in Fig. 3 for easier qualitative

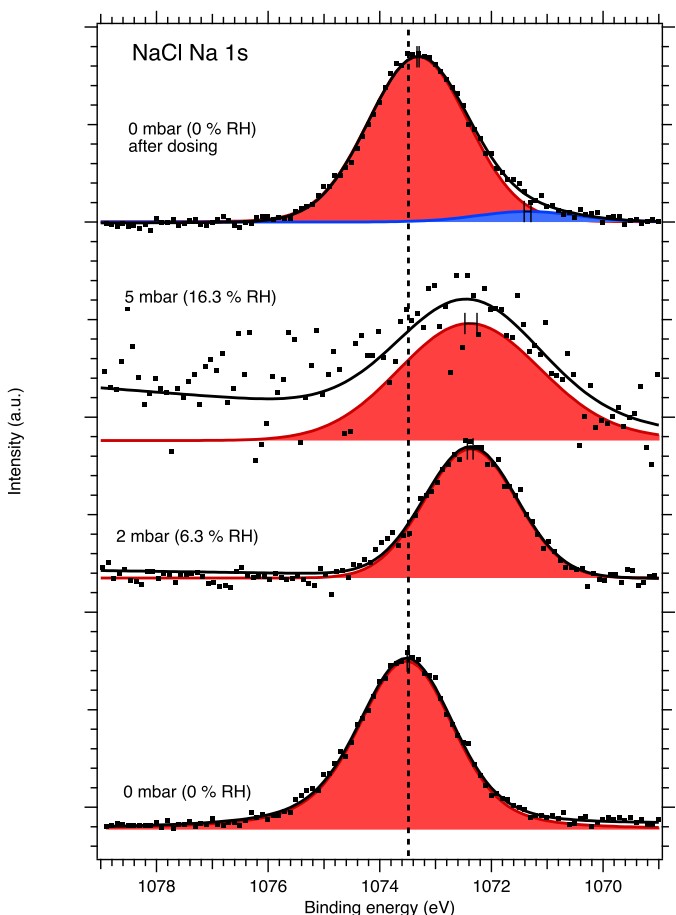

**Figure 1.** Na 1s XPS spectra of NaCl aerosol particles recorded at different water vapor pressures (relative humidities, RH). The point markers show the recorded data and the solid lines the fit envelope curve. The spectra are fit using a single, symmetric Voigt peak shown in red with an additional peak in blue necessary to explain the spectrum at 0 mbar after water exposure. The dashed vertical line shows the binding energy of Na 1s at 0 mbar pressure (0 % RH) at the beginning of the experiments. Error bars show the estimated uncertainty in the peak position from Monte Carlo analysis. Photon energy was 1486.6 eV from the Al anode.

comparison of the two data sets. The shift of the Na 1s peak for deposited aerosol in the present work towards larger binding energies at 16 % RH is not entirely in line with the observations on NaCl (001), which show monotonic shifts towards lower binding energies both for Na 2s and Cl 2p peaks. However, the poorer quality of the experimental spectrum for aerosol Na 1s at 16 % RH in the present work, compared to spectra recorded at lower RH, leads to larger uncertainty in the peak fitting procedure. Furthermore, the shifts observed for NaCl nanoparticles in the present work are larger for Cl 2p than for Na 1s, as opposed to larger shifts (by approximately 50 meV) for Na 2s than Cl 2p in the NaCl (001) single crystal.

In our experiments, the water vapor was removed from the AP cell after the measurements at 16 % RH and another set of spectra was recorded at 0 % RH. We see that when RH decreases from 16 % to 0 %, the binding energies of the Na 1s and Cl

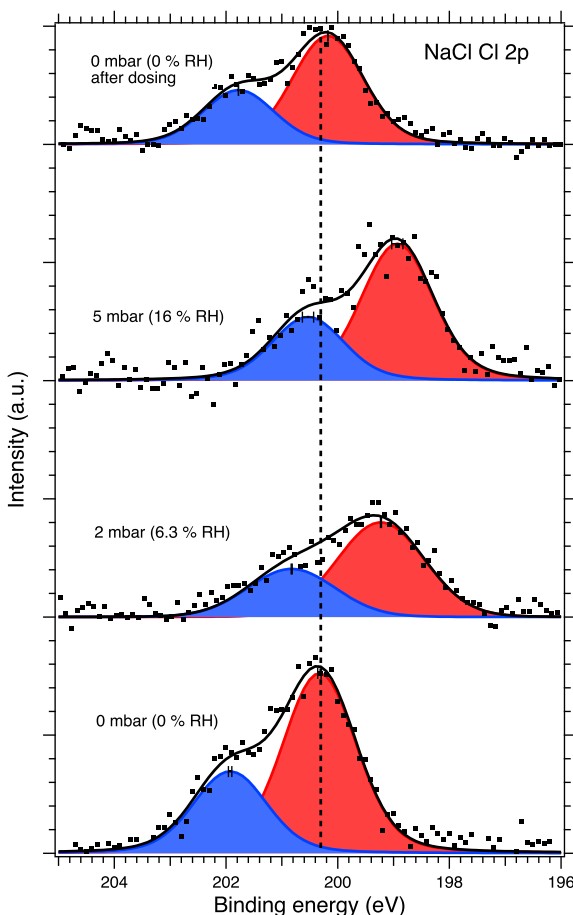

**Figure 2.** Cl 2p XPS spectra of NaCl aerosol particles recorded at different water vapor pressures (relative humidities, RH). The dots show the recorded data, the solid lines the fit envelope curve and the red and blue peaks the fitted 3/2 and 1/2 spin-orbit components, respectively. The dashed vertical line shows the binding energy of Cl 2p 3/2 component at 0 mbar pressure (0 % RH) at the beginning of the experiments. Error bars show the estimated uncertainty in the peak position from Monte Carlo analysis. Photon energy was 1486.6 eV from the Al anode.

2p do not shift back to their original values, but a memory effect of approximately 0.2 eV for both Na 1s and Cl 2p is observed. Since this observed memory effect is greater than the energy step size of 0.1 eV used in the acquisition of the spectra which is in turn greater than the energy accuracy of the electron analyzer when operating at 50 or 100 eV pass energy, we consider it to be a real effect that cannot be explained solely by experimental uncertainty. This is in line with the observations by Verdaguer et al. (2008) who also report a memory effect of less than 0.5 eV for single crystal NaCl (001).

### 3.1.2 Changes in spectral peak width

In addition to the shifts in binding energies, we also investigated possible changes in the width of the XPS peaks due to the effects of water adsorption to the particle surfaces. Changes in the spectral peak widths can indicate changes in the chemical

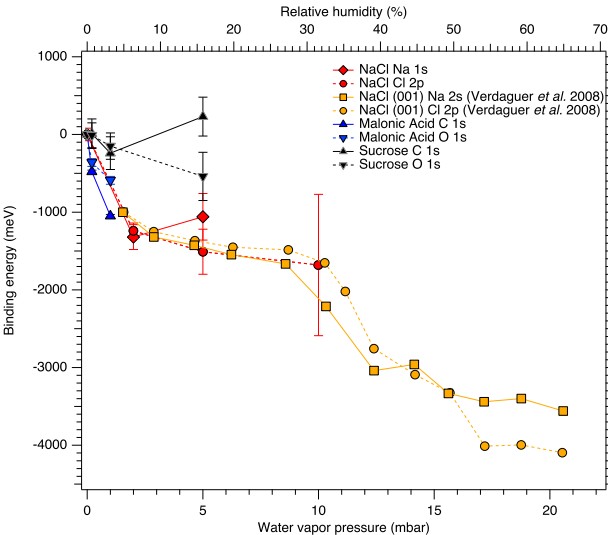

**Figure 3.** Binding energy shifts relative to dry conditions of the core level peaks for NaCl, sucrose and malonic acid particles, as functions of water vapor pressure (relative humidity). The orange traces extending above 60 % RH show the data from Verdaguer et al. (2008) for single NaCl (001) crystal that has been aligned to the shifts obtained in this study.

environment or charge carrier mobility on the aerosol surfaces. While we found clear binding energy shifts (Fig. 3), we did not observe any significant changes in the peak width of either the Na 1s or the Cl 2p peaks. In contrast, the data obtained by Verdaguer et al. (2008) show decreasing total widths of Na 2s and Cl 2p peaks for NaCl (001) single crystal with increasing relative humidity, qualitatively following the behavior of their observed core level binding energy shifts.

The difference between deposited aerosols and single crystal spectra may, at least partially, be explained by the experimental
resolution, which is lower in the present study with peak widths dominated by experimental factors. In the NaCl (001) case, the observed binding energy shifts are attributed to the discharging of the surface, where increasing relative humidity causes the surface to be gradually discharged due to ionized water vapor and secondary electrons. As water is adsorbed onto the surface, it leads to solvation of surface ions and increased mobility of these charge carriers. This ion mobility makes discharging more efficient, leading to further shifts in observed binding energies. According to Verdaguer et al. (2008) these solvation effects
become more significant above approximately 35 % RH.

This interpretation is further supported by scanning force microscopy studies and infrared studies that show modifications in the surface structure and sudden increase of water coverage at around 40 % RH (Dai et al., 1997; Peters et al., 1997; Foster and Ewing, 2000). Kelvin probe microscopy (KPM) experiments by Cabrera-Sanfelix et al. (2007), Verdaguer et al. (2005), and Verdaguer et al. (2008) show how water vapor affects the surface potential of the NaCl (001) surface. They found variations on
the order of 0.1 V to 0.25 V across the RH range of their experiments, which clearly is not sufficient to explain the energy shifts they observed. The variations in surface potential also cause peak broadening due to slight differences to the kinetic energies of photoelectrons from different regions of the surface. The decrease in peak broadening observed for single crystals is ultimately

attributed to adsorbed water reducing inhomogeneities in the surface potential. These inhomogeneities originate from potential differences between and within the step and terrace sites of the crystal. The KPM experiments show that the inhomogeneities are removed as water adsorbs at very low surface coverage below that of one monolayer. In the present work, the deposited NaCl aerosol particle samples are most likely very far from being perfect crystals, possibly containing a large number of steps, terraces and kinks leading to larger variations in the surface potential and thus increased broadening of the peaks in any humidity condition compared to a NaCl (001) crystal. This could, together with the moderate experimental resolution, explain why no decrease in the broadening of the peaks is observed for the aerosol particles at low RH.

Results of our study show, not surprisingly, that the aerosol particle samples investigated have more complex morphology than the simple single crystal surfaces previously studied by XPS. Several previous studies have observed that the process of drying an aerosol can indeed affect its crystalline form. For example, studies of NaCl aerosol particles generated from drying of aqueous droplets have inferred a non-crystalline structure with pores or pockets that trap liquid water (Weis and Ewing, 1999; Darr et al., 2014; Braun and Krieger, 2001). This is explained by the presence of liquid water detected below the deliquescence relative humidity but still at much higher RH than in our study. Furthermore, the morphology of NaCl particles expressed via the shape factor has been shown to be controlled by the drying rate (Wang et al., 2010). A recent study (Archer et al., 2020) has explained the morphology of particles formed from drying of a colloid as a competition between diffusion of solute in solution versus loss of solvent, with higher solvent loss rate compared to solute diffusivity leading to more complex morphologies. For atmospheric samples, microscopy studies on sea salt particles have shown them to have complex morphologies (e.g. Cheng et al., 1988), similarly to what was found for the laboratory generated aerosol samples in the present study.

Atmospheric aerosols are likely to undergo multiple drying and humidification cycles under a wide range of conditions and thus to exhibit a range of morphologies related to drying. Our measurements on aerosol particle samples generated from nebulization and subsequent dessication are therefore expected to much more closely represent the morphologies of atmospheric aerosols, compared to the simple uniform morphologies of single-crystal samples.

### 3.1.3 Changes in peak area ratios: chemical changes in surface composition

Considering the relative amounts of Na and Cl on the particle surfaces, we observe a decrease in the Na 1s to Cl 2p peak area ratio, even after taking differences in photoionization cross section and attenuation due to scattering in the water vapor into account. This change as a function of increasing RH is presented in Fig. 4. The extracted Na to Cl ratio of 0.68 at 0 % RH and decreases slightly to 0.51 and 0.58 when the RH is increased to 6.3 % and 16 %, respectively. The ratio measured at 0 % RH after dosing was determined to be 0.62 which is close to the value before exposure to water vapor. As the spectra for all RH are recorded with a constant excitation energy from the Al K$\alpha$ anode, different transmission through the electron analyzer at different electron kinetic energies cannot explain the difference between the Na to Cl peak area ratio and the NaCl stoichiometric ratio. Due to the constant relative transmission, this could lead only to a constant ratio differing from the stoichiometric ratio. Furthermore, the reported transmission of the analyzer is very nearly the same for the Cl 2p and Na 1s regions (SPECS Surface Nano Analysis GmbH). This correlation between changes in the chemical composition of the surface layer and RH is therefore a clear indication of water being adsorbed to the NaCl particle surfaces. Assuming that

the thickness of the layer of water molecules adsorbed on the NaCl surfaces is approximately 2.4 Å (Cabrera-Sanfelix et al., 2007), photoelectrons emitted from the NaCl particles need to pass through this distance to escape from the sample and would be attenuated by collisions with the water molecules in the process. In our experimental conditions, the photoelectrons have kinetic energies (rounded to the nearest ten) of approximately 410 eV and 1290 eV for Na 1s and Cl 2p, respectively. The two core level signals will therefore be attenuated differently due to the different inelastic mean free paths of the electrons, approximately 2 nm and 5 nm, respectively (Emfietzoglou and Moscovitch, 2002).

To quantify the attenuation of the photoelectron signal, we use an exponential decay function $I_n = I_n^0 e^{-x/\lambda_n}$, where $I_n$ is the attenuated intensity of peak $n$, $I_n^0$ is the corresponding unattenuated intensity, $x$ is depth into the sample from where the signal originates, and $\lambda_n$ the energy-dependent inelastic mean free path of photoelectrons contributing to peak $n$. The depth of origin can be expressed as $x = \frac{\lambda_1 \lambda_2}{\lambda_1 - \lambda_2} \ln R$, where $R = \frac{I_1 I_2^0}{I_2 I_1^0}$ is the relative ratio of attenuated and unattenuated signals from two separate peaks $n = 1, 2$. We here use the total integrated peak areas to represent signal intensities. In our experiments, the unattenuated signal ratio (measured without water vapor) for Na 1s and Cl 2p is $I_{\mathrm{Na}}^0 / I_{\mathrm{Cl}}^0 = 0.9$. With this, from the corresponding signal ratios at elevated humidities, the simple attenuation model gives depths of photoelectron origin (or water layer thickness) of approximately 14 and 4 Å for 6.3 and 16 % RH, respectively. This is much more than the previous observations of 2.4 Å by Cabrera-Sanfelix et al. (2007) and also counterintuitive as it would mean decreasing layer thickness with increasing RH. Furthermore, this simple attenuation model implicitly assumes at least a full monolayer coverage so that all signal is uniformly attenuated by the adsorbed water layer. Previous studies on NaCl (100) crystals (Peters et al., 1997; Peters and Ewing, 1997; Foster and Ewing, 2000) have shown that a full water monolayer does not develop until approximately 35 % RH. On the other hand, it has been shown that NaCl (100) does not adsorb water strongly whereas small particles (1-10 $\mu$m) have a propensity for adsorbing large amounts of water (Ghosal and Hemminger, 2004). This strong adsorption has also been connected to observations of water remaining on the particle surface, even at elevated temperatures in vacuum. Indeed, this is also observed in the present work in the form of a memory effect in the Na 1s and Cl 2p peak shifts as the humidity is decreased to 0 % RH: the peak shift does not completely disappear, even after heating the sample to 125 °C (Fig. S1 in the supplement). The signs of stronger water adsorption in the present work could therefore be attributed to the small size of our sampled NaCl particles, contributing to larger surface area (as well as more crystal imperfections) and therefore to a larger number of possible surface adsorption sites.

While attenuation observed due to the adsorbed water layer is one possible explanation for the decreasing Na to Cl ratio observed in this work, another, complementary, explanation could be the preferential dissolution of Cl$^-$ ions into the water film. Spatial segregation of the Na$^+$ and Cl$^-$ ions could further enhance the difference in the attenuation between Na 1s and Cl 2p XPS signals and is likely to be further enhanced as a more complete water layer forms on the particle surfaces. In this case, the Cl 2p photoelectrons only need to pass through part of the water layer while the Na 1s electrons will be attenuated by both a layer of water as well as the dissolved Cl$^-$ ions. This interpretation is supported by density functional theory (DFT) calculations and contact potential measurements (Cabrera-Sanfelix et al., 2007), which showed that Cl$^-$ can indeed be lifted out of a NaCl (100) crystal surface already at one monolayer water coverage. Using XPS on free-flying sub-2 nm CsBr water clusters, Hautala et al. (2017b) found that Br$^-$ ions were closer to the surface than their counter cations. These experimental

results further support the interpretation of our present observations that pre-deliquescent water adsorption enhances chloride relative to sodium in the aerosol surface layer.

The presence of halide ions, especially $Cl^-$ and $Br^-$, at the air-water interface has been connected to increased photochemical activity (e.g. George and Abbatt, 2010, and references therein). The mobilization of ions can lead to release of gaseous halogen compounds from sea salt aerosol due to reactions in the aqueous phase (Mozurkewich, 1995; Vogt et al., 1996; Kerminen et al., 1998; Keene et al., 1999). In the atmosphere, formation of solvated halogen ions even at very dry conditions via similar pre-deliquescent adsorption of water onto the surfaces of sea salt aerosol as seen for laboratory generated aerosol in this work could therefore have significant implications for the halogen cycle, including ozone chemistry in the polar regions (Simpson et al., 2007).

Another possible explanation for the difference in the Na to Cl ratio could be beam damage. Cl has been shown to be more prone to the effect of irradiation, and in the course of the experiments this could mean an accumulating loss of Cl due to desorption or diffusion (Verdaguer et al., 2008). Our present results, however, show exactly the opposite: the Cl is enhanced with respect to Na. Furthermore, the radiation from the X-ray anode is far less intense than that produced by a synchrotron, making the effects due to beam damage less likely. This leads us to conclude that the observed effect cannot be due to beam damage. Finally, it is also possible that the photoelectron signal is attenuated by so-called adventitious carbon, a thin layer of carbon-containing material that may form on the samples when exposed to ambient air (Barr and Seal, 1995), in addition to adsorbed water. During our experiments, we unfortunately did not measure C 1s spectra for the NaCl particles and we are therefore not able to quantify the potential influence of adventitious carbon on the Na 1s and Cl 2p peak area ratios. Effects of adventitious carbon is discussed in more detail for the sucrose and malonic acid samples where C 1s spectra were measured.

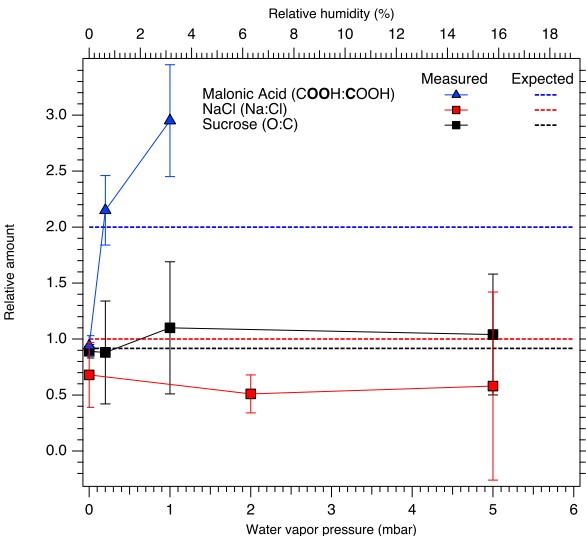

**Figure 4.** Relative peak areas for NaCl (Na:Cl, red), sucrose (O:C, black) and the malonic acid carboxyl groups (**COOH:C**OOH, blue). The dashed horizontal red, black and blue lines show the expected stoichiometric ratio for NaCl, sucrose and malonic acid, respectively. Relative ratios differing from the stoichiometric ratio indicate changes in the chemical composition of the surface of the aerosol particles.

## 3.2 Sucrose

Figure 5 shows the C 1s XPS spectra of deposited sucrose aerosol particles at 0, 0.2, 1 and 5 mbar water vapor pressure. Measurements were repeated at 0 mbar water vapor pressure after dosing but are not shown here since no significant changes were observed. The C 1s spectra were fitted with three peaks corresponding to distinct carbon atoms in C-C/C-H, C-O and O-C-O bonds at 285.90, 287.36 and 288.76 eV, respectively, at 0 % RH. Sucrose molecules contain only C-O and O-C-O bound carbon and the largest peak corresponding to C-C/C-H bound carbon is therefore attributed to adventitious carbon on the substrate. The obtained peak energies at 0 % RH for C-O and O-C-O carbons are in good agreement with values reported by Stevens and Schroeder (2009), 286.7 and 288.1 eV, respectively. Since the gold substrate is highly conductive, we do not expect any charging effect as for the NaCl samples deposited on Si. Differences between the binding energies measured here and those of Stevens and Schroeder (2009) may instead arise from the choice of reference for calibrating the energy scale. Whereas we here calibrated the energy scale using the well-known Au 4f peak, Stevens and Schroeder (2009) used the adventitious carbon peak, which may have a wide range of binding energies (Greczynski and Hultman, 2018). Additional details on the peak fitting can be found in the supplement.

Figure 6 shows the O 1s XPS spectra of the deposited sucrose aerosol particles. The O 1s spectra were fitted with two peaks for sucrose corresponding to C-O and O-C-O bound oxygen. With the introduction of water vapor into the high pressure cell, a third peak from the wator vapor (Patel et al., 2019) appears in the XPS spectra, as evidenced by the increase in signal with increasing humidity. The binding energies of the C-O and O-C-O peaks at 0 % RH were determined to be 533.02 eV and 533.72, respectively. These values match closely with those (533.0 eV and 533.7 eV, respectively) reported by Stevens and Schroeder (2009). Additional peak fitting details can be found in the supplement.

The binding energy shifts of sucrose C 1s and O 1s peaks with respect to 0 % RH are shown in the black traces in Fig. 3. Because the peak separation in each case is fixed, only a single shift is reported for spectra of each element. Unlike with NaCl, the binding energy shifts in the sucrose spectra do not suggest adsorption of water to the sucrose particles at the investigated humidities. The C 1s shows initially a small shift and then returns to 0 at 16 % RH. The O 1s peaks on the other hand seem to shift slightly towards lower binding energies (580 meV at 16 % RH). Figure 4 shows the O:C ratio of sucrose as a function of RH. The ratio is in all conditions very close to the stoichiometric value of O:C$= 11/12 \approx 0.92$. Looking specifically at the relative ratios of C-O and O-C-O bound carbon C 1s and oxygen O 1s peaks, we see only minor changes with RH. The C-O C 1s signal seems to increase slightly from 0 % to 3.2 % RH, but is again lower at 16 % RH. The same is seen for the O 1s O-C-O signal, which increases slightly from dry conditions towards 16 % RH. Taken together, the observations that both the O 1s and C 1s peaks only show a small binding energy shift with increasing RH, the O:C ratio remains nearly constant and the changes in relative ratios between C-O and O-C-O bound carbon C 1s and oxygen O 1s signals are small, suggest that either there is no significant amounts of water adsorbed onto the particles at RH below 16 % or that, if water is adsorbed, there is no significant chemical interaction between the water and the sucrose nanoparticle.

The high surface sensitivity of the XPS spectra enables us to confirm previous measurement results that rely on detecting changes in volume (Zobrist et al., 2011) or mass (Yao et al., 2011). Zobrist et al. (2011) used optical techniques to determine the

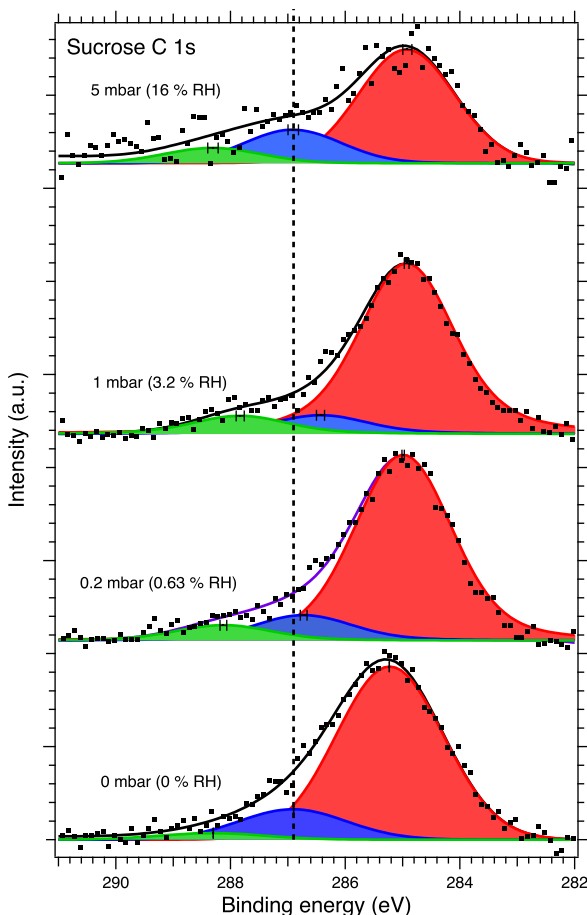

**Figure 5.** C 1s XPS spectra of sucrose aerosol particles. The dots show the recorded data, the solid lines the fit envelope curve and the red, blue and green peaks the C-C/C-H, C-O and O-C-O bound carbon components, respectively. The dashed vertical line shows the binding energy of the C-O component at 0 mbar pressure (0 % RH) at the beginning of the experiments. Error bars show the estimated uncertainty in the peak position from Monte Carlo analysis. Photon energy was 1253.6 eV from the Mg anode.

size of single sucrose microparticles levitated in an electrodynamic balance (EDB) under varying RH conditions. Their results show that crystalline sucrose barely adsorbs any water before the deliquescence point of 85.6 %. They also show that sucrose does not exhibit any efflorescence upon drying, but instead assumes an amorphous glassy state. On subsequent humidification cycles the amorphous sucrose particle did not show any significant water uptake before RH of 35-40 %. Our present work also supports this observation on the molecular level. In contrast to NaCl particles, the absence of changes in the sucrose C 1s and O 1s binding energies and O to C ratio with increasing RH, together with this high surface sensitivity of the XPS measurements, indicate insignificant water adsorption by sucrose aerosol particles at humidities up to 16 % RH.

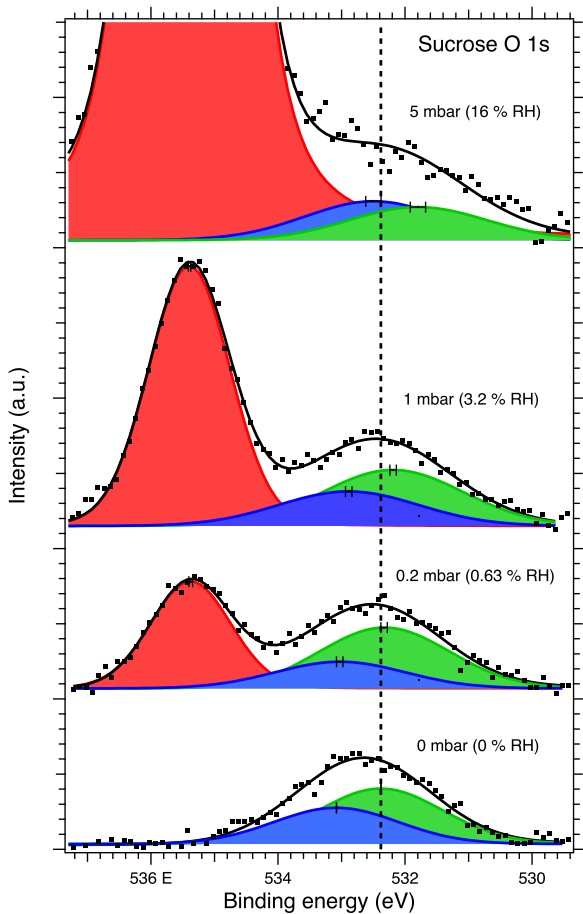

**Figure 6.** O 1s XPS spectra of sucrose aerosol particles. The dots show the recorded data, the solid lines the fit envelope curve and the red, blue and green peaks the water vapor, and sucrose O-C-O and C-O components, respectively. The dashed vertical line shows the binding energy of the C-O component at 0 mbar pressure (0 % RH) at the beginning of the experiments. Error bars show the estimated uncertainty in the peak position from Monte Carlo analysis. Photon energy was 1253.6 eV from the Mg anode.

In the atmosphere, these results could mean that pre-deliquescent water adsorption and resulting chemical changes in very dry conditions could be much less likely to occur for aerosol comprising significant amounts of sugar functionalities, such as those produced in biomass burning (Scaramboni et al., 2015; Bhattarai et al., 2019), compared to sea salt aerosol.

## 3.3  Malonic Acid

Figures 7 and 8 show the recorded C 1s and O 1s XPS spectra, respectively, of deposited malonic acid aerosol particles at 0, 0.2, and 1 mbar water vapor. Due to measurement time constraints, spectra were not recorded for 5 mbar or again at 0 mbar water vapor after dosing. The C 1s spectra were fitted with three main peaks corresponding to C-C/C-H bound carbon originating from adventitious carbon, and COOH and C-C carbon from malonic acid particles at 285.75, 290.02 and 286.02

eV, respectively, in UHV conditions. As for sucrose, we also do not expect any charging effects for the malonic acid samples due to the high conductivity of the gold substrate. Malonic acid has been shown to be prone to beam damage when exposed to radiation from laboratory X-ray sources. This can give rise to additional peaks in the XPS spectra arising from malonic acid molecules damaged by the X-ray beam (Ferreira Jr. et al., 2017a, b). The decomposition peaks DP1 and DP2 observed by Ferreira Jr. et al. (2017b) were included in the fit for 0 % RH (UHV), where a good fit could not be produced without their inclusion. In UHV conditions, the DP1 and DP2 peaks were observed at 291.52 eV and 287.22 eV. The spectra recorded with water vapor did not show clear indication of the DP2 feature, but an additional peak needs to be fitted close to the DP1 feature in both 0.6 % and 3.2 % RH spectra. The best fit was acquired with the feature at 287.05 eV and 286.50 eV for 0.6 % and 3.2 % RH, respectively. This is 1.4 eV higher than the C-C carbon peak of malonic acid in both cases, in good agreement with the position of the DP1 peak of Ferreira Jr. et al. (2017b). Additional details of the fitting procedure may be found in the supplement.

The malonic acid O 1s spectra were fitted with two main peaks for the C=O and C-OH bound oxygen of the two carboxyl groups. For UHV conditions, the best fit was acquired with a splitting of approximately 1.0 eV between the C=O and C-OH peaks (at 532.33 and 533.28 eV, respectively). This splitting is smaller than the previously reported values of 1.1 and 1.3 eV by Ferreira Jr. et al. (2017b). For our measurements with water vapor, the best fits were obtained with a splitting of 1.2 eV between the C=O and C-OH oxygen peaks (at 531.84 and 533.04 eV at 3.2 % RH, respectively), now in good agreement with the work of Ferreira Jr. et al. (2017b). The peak at highest binding energy in the spectra recorded at humid conditions is the O 1s peak from water vapor. While peaks due to possible beam damage were observed in the C 1s spectra of malonic acid (DP1 and DP2 in Fig. 7), fitting corresponding peaks in the O 1s spectrum did not yield a better representation of the measured XPS spectra. However, this is also in line with the previous works of Ferreira Jr. et al. (2017a), showing that the signal due to beam damage is much less pronounced in the O 1s spectra than in the C 1s spectra.

The binding energy shifts of malonic acid particle C 1s and O 1s peaks with respect to 0 % RH are shown in the blue traces in Fig. 3. When RH is increased, the COOH (and thus also C-C) peaks of malonic acid are seen to shift towards lower binding energies. While the the shift for malonic acid O 1s is not as dramatic as for NaCl particles, the malonic acid C 1s binding energies shift is even more pronounced than that of Na 1s or Cl 2p, indicating that water is being adsorbed onto the surface of the malonic acid particles and changing the chemical environment of the surface molecules. In addition to the peaks shifts, the O 1s to C 1s signal ratio within the malonic acid carboxyl group, **CO**O**H:C**OO**H**, changes with increasing RH. We focus on the carboxyl groups as the C-C/C-H contribution is under the strong C-C/C-H signal from adventitious carbon and therefore cannot be fitted reliably. The stoichiometric O:C ratio of a carboxyl group is 2. Interestingly, already at 0 % RH (UHV), the extracted ratio **CO**O**H:C**OO**H**$= 0.94$ is far from this stoichiometric value. When introducing water vapor into the system at 0.6 % RH, the ratio changes to 2.15. At 3 % RH, it increases even further to 2.95. While we cannot offer a clear explanation for this observation at the present, based on the observed binding energy shifts and the changes in the **CO**O**H:C**OO**H** ratio, it is safe to conclude that water is adsorbed on nanoscale malonic acid aerosols particles already at the lowest RH measured and that it has an effect on the chemical composition of the particle surface.

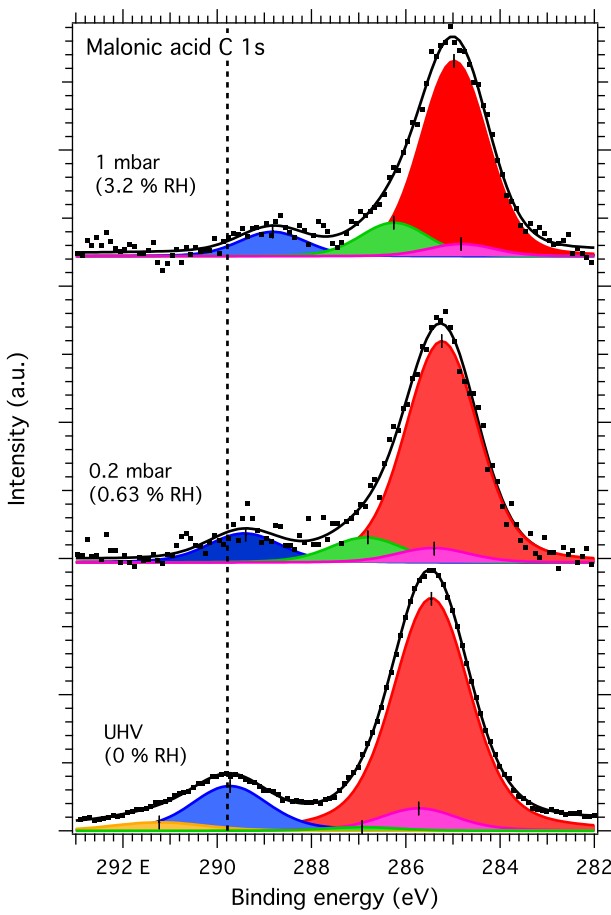

**Figure 7.** C 1s spectra of malonic acid aerosol particles. The dots show the recorded data and the solid lines the fit to the data. The blue and purple peaks correspond to COOH and C-C of malonic acid, respectively while the yellow and green peaks correspond to DP1 and DP2 components, respectively, attributed to beam damage. The red peak corresponds to C-C/C-H of adventitious carbon. The dashed vertical line shows the binding energy of the COOH component at 0 mbar pressure (0 % RH) at the beginning of the experiments. Error bars show the estimated uncertainty in the peak position from Monte Carlo analysis. Photon energy was 1253.6 eV from the Mg anode.

On closer inspection of the malonic acid spectra, it is clear that the C 1s spectrum at 0 % RH (UHV) resembles that of a
450   beam damaged sample described by Ferreira Jr. et al. (2017a, b), but not entirely: the DP2 feature in our particle spectra is stronger than reported by Ferreira Jr. et al. (2017b) and the DP1 weaker. When increasing the RH in our measurements, the stronger DP2 peak disappears and only the extra feature attributed to DP1 is seen. Our malonic acid particle O 1s spectra do not show any signs of beam damage even at increased RH. Instead, the relative intensities between the C=O and C-OH components change toward the expected C=O:C-OH ratio of 1 from the C=O:C-OH = 1.4 observed at 0 % RH (UHV). This indicates an
455   increase in the number of C-OH and decrease in the number of C=O groups on the malonic acid particle surface with increasing RH. Indeed, the binding energy of the C 1s (DP2) feature is close to the typical binding energy (approximately 286.5 eV) of a

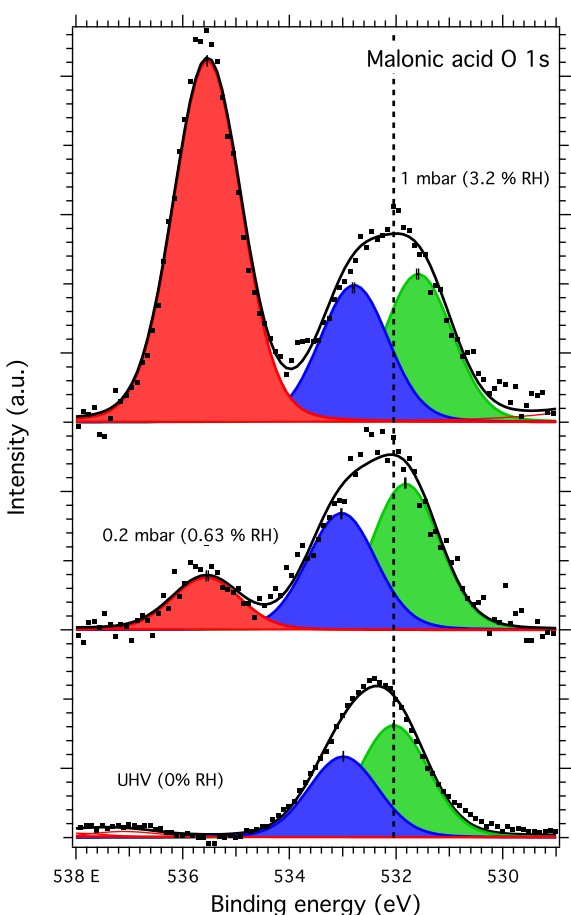

**Figure 8.** O 1s spectra of malonic acid aerosol particles. The dots show the recorded data, the solid lines the fit envelope curve and the red, blue and green peaks the water vapor, C-OH and C=O components, respectively. The dashed vertical line shows the binding energy of the C=O component at 0 mbar pressure (0 % RH) at the beginning of the experiments. Error bars show the estimated uncertainty in the peak position from Monte Carlo analysis. Photon energy was 1253.6 eV from the Mg anode.

hydroxyl group carbon, C-OH. If this were purely a result of beam damage, the DP2 feature in the C 1s spectrum should also be visible, not to mention the beam damage component in the O 1s spectra. It could therefore be a sign of significant chemical changes in the malonic acid particles.

460    Malonic acid is known to exhibit keto-enol tautomerism and it has been theoretically predicted that the transition between the two forms can take place through an intramolecular transition in a system of just six water molecules and one malonic acid molecule (see e.g. Yamabe et al. (2004)). Ghorai et al. (2011) performed spectromicroscopic experiments combining scanning transmission X-ray microscopy with near-edge X-ray absorption fine structure spectroscopy (STXM/NEXAFS) and Fourier transform infrared spectroscopy (FTIR) on deliquesced submicron malonic acid aerosol particles at varying RH up to 90 %.

465    Their bulk-sensitive experiments show that while the keto form dominates in dilute aqueous solutions of malonic acid, the

equilibrium shifts toward the enol form in saturated solutions. This observation is also supported by the theoretical calculations of Dick-Pérez and Windus (2017) for concentrated malonic acid solution particles. The results of Ghorai et al. (2011) show that the amount of enol form increases with increasing relative humidity as water is absorbed by deliquesced malonic acid aerosols. Furthermore, they demonstrate water uptake by malonic acid aerosols already at RH as low as 2 %, which is in good agreement with the conclusion of our surface-sensitive XPS experiments in the present work.

The single particle carbon K-edge NEXAFS spectra of Ghorai et al. (2011) show a clear increase in C=C and C-OH resonances, while C-COOH and C-CH are decreased with increasing RH. Their oxygen K-edge spectra also shows a clear decrease in the -COOH resonance. This is similar to the observations of the present work, where the O 1s C-OH signal from malonic acid particle surfaces increases with increasing RH. While a similar increase is also observed here in the C 1s for the C-OH (DP1) component, caution is necessary when attributing this to the enol form of malonic acid. The C-OH peak at approximately 286.5 eV is a sign of carbon with a single hydroxyl group, while the malonic acid enol form would have a carbon with two hydroxyl groups bound to it (HO-C-OH or O-C-O), increasing the binding energy from 286.5 eV to approximately 288 eV (Lannon Jr. and Meng, 1999). Although no such peak has been fit to the data in the present work, it also cannot be ruled out completely. However, if the spectral changes observed in our XPS experiments are due to the formation of the enol tautomer on the malonic acid particle surface, this could have profound implications in terms of atmospheric chemistry as the keto and enol forms have differing physical and chemical stabilities and reaction pathways in both the gas and aqueous phases (Yamabe et al., 2004).

## 4    Conclusions

The main goal of this work has been to investigate the potential for obtaining meaningful results with the APXPS technique for aerosol samples comprising simple but atmospherically relevant chemical components. We studied water adsorption onto NaCl, sucrose and malonic acid aerosol particles deposited on silicon and gold substrates using the APXPS with a laboratory X-ray source (Al K$\alpha$ and Mg K$\alpha$). To our knowledge, this is the first time surface sensitive and chemically specific XPS has been used in AP conditions on nano-scale particles with compositions of immediate atmospheric relevance.

The samples were exposed to relative humidities between 0 % and 16 % at room temperature and the XPS spectra of Na 1s, Cl 2p, C 1s and O 1s core levels were recorded. The results show that water can be adsorbed onto the aerosol particles already at very low relative humidities. Corresponding changes were observed in the chemical environment, and possibly even composition, of the particle surfaces with increasing relative humidity. For NaCl particles, we find signs of Cl$^-$ ions dissolving into the thin surface water layer, as well as a memory effect in binding energies when drying samples back to 0 % RH. This water uptake memory persists even after heating the particles up to 125 °C. Although some water seems to remain on the particles, the Cl appears to return to the crystal lattice of the aerosol as the particle is dried. Sucrose particles show very few changes when exposed to water vapor at the conditions of these experiments. Malonic acid particles on the other hand show a dramatic shift in the C 1s binding energies even when exposed to very small amounts of water (0.2–3 % RH), as well as a shift in the O 1s binding energies. Changes are also seen in the relative amounts of C=O and C-OH bound oxygen on the surface of

malonic acid particles, together with the emergence of C-OH bound carbon at the expense of COOH. These surface specific changes are in line with earlier bulk-sensitive observations of water uptake by malonic acid particles at very low RH (Ghorai et al., 2011) and could even indicate a change in the keto-enol equilibrium of malonic acid within the surface adsorbed water layer in our experiments.

The present observations suggest that while sucrose particles do not interact appreciably with water vapor at low RH conditions, the adsorption of small amounts of water onto both NaCl and malonic acid particles could already have implications for their surface chemistry on a molecular level. These differences between chemically simple components emulating sea salt, biomass burning, and carboxylic acid aerosol could represent major differences in their atmospheric chemistry in response to water in dry conditions. The presence of water at the aerosol surface could impact water-catalyzed chemical reactions, as well as reactions involving mobilization of charged species such as organic and inorganic ions or components with acid-base functionalities. For example, $Cl^-$ ions in sea salt aerosols are a source of chlorine in the troposphere through heterogeneous processes (see e.g. Wang et al. (2019) and references therein). With sea salt being one of the most abundant inorganic aerosol components in the global atmosphere, our present observations could imply significant reactive uptake and processing of atmospheric trace gases even at low RH. Dicarboxylic acids like malonic acid are also ubiquitous in atmospheric aerosols in the troposphere. The formation of concentrated solutions with the enol form present even in relatively dry conditions may have important implications for the reactivity of these aerosol surfaces. The role of particle surfaces is enhanced for smaller particles with large surface-to-bulk ratios (Prisle et al., 2010b; Bzdek et al., 2020). The effects of water uptake detected at very low relative humidities in this work could not have been detected with standard mass or volume based aerosol techniques, nor attributed specifically to the surface with bulk sensitive spectroscopic methods.

Our results demonstrate the viability of the APXPS technique for *in situ* studies of vapor interaction with aerosol particle surfaces. XPS can provide surface-specific, molecular-level chemical information on uptake and reactions through changes in the presence, position or intensity of spectral peaks, and can detect chemical changes on aerosol surfaces which likely cannot be resolved with even high-resolution bulk methods such as aerosol mass spectrometry. We have shown that APXPS can be used on sampled aerosol particles comprising both organic and inorganic components of high atmospheric relevance. Our samples were generated with standard aerosol laboratory methods and exposed to a well-defined vapor phase during experiments. The present experiments were carried out using a conventional laboratory X-ray source which limited the ambient pressure at which the measurements could be done. While the experiments were conducted at atmospherically relevant water vapor pressures, they were still far below the deliquescent relative humidities of the compounds studied and we cannot immediately conclude that the interaction between water vapor and the nanoparticles studied here will remain the same for all relative humidities below deliquescence. Combining APXPS with high brilliance synchrotron radiation now available at modern synchrotron facilities therefore shows great promise for studying aerosol surfaces at even higher ambient pressure conditions up to deliquescence, in more complex and reactive environments, as well as with high time resolution.

*Author contributions.* Conceptualization: NLP; Funding acquisition: NLP; Investigation: JJL, KRR, SW, EK, M-HM, and SU; Analysis: SU, JJL, SW, and NLP; Visualization: SU and JJL; Writing of first version: JJL, SU, and NLP with input from all co-authors. Writing of revised version: JJL and NLP with input from all co-authors. Authors' response: JJL, NLP, and SU with input from all co-authors; Supervision: NLP; Project administration: NLP.

535   *Competing interests.* The authors declare that they have no conflict of interest.

*Acknowledgements.* This project has received funding from the European Research Council (ERC) under the European Union's Horizon 2020 research and innovation programme, Project SURFACE (Grant Agreement No. 717022). The authors also gratefully acknowledge the financial contribution from the Academy of Finland, including Grant Nos. 308238, 314175, 335649, 290145, 326291, and 331532. We acknowledge MAX IV Laboratory for time on Beamline SPECIES. Research conducted at MAX IV, a Swedish national user facility, is
540   supported by the Swedish Research council under contract 2018-07152, the Swedish Governmental Agency for Innovation Systems under contract 2018-04969, and Formas under contract 2019-02496. We warmly thank Jenny Rissler and Birgitta Svenningsson for assistance with preparation of samples and estimation of surface coverage.

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
