# Peer review of "Pre-deliquescent water uptake in deposited nanoparticles observed with *in situ* ambient pressure X-ray photoelectron spectroscopy"

_Atmospheric Chemistry and Physics, 2020_

## Referee Comment (RC1) · Anonymous Referee #1 · 13 May 2020

This is an interesting article describing the application of a new technique to study pre-deliquescence water uptake and the changes in the chemical environment near surface based on changes of XPS peak characteristics of model aerosol compounds as a function of relative humidity. The work is novel. The paper is heavy on discussing spectral characteristics and less on atmospheric implications. Much of the discussions rely on spectral fitting of the spectra, which at times are quite noisy. In general, the authors need to provide more information on how uncertainties due to the experiments as well as fittings would affect the results.

In the Introduction, the authors did a good job in identifying the need of surface analysis

as well as pre-deliquescence water uptake. However, they seem to have missed some useful literature relevant to this paper. 1. XPS has been used to examine ambient particles collected by a MOUDI. Cheng et al. (2013, 2014) analyzed particles from various stages, including the last stage of 0.056 to 0.1 microns. 2. Gen et al. (2017, 2019) have used SERS to detect the presence of surface adsorbed water and the formation of BrC on particle surface. 3. Pre deliquescence water adsorption/absorption of NaCl has been indirectly discussed in HTDMA measurements, albeit at much higher RH than the range studied in this paper. I also encourage the authors to look into the literature of the changes of the shape factors as indication of pre-deliquescence water uptake.

Line 148: The authors discussed the shift in binding energy and the intensities of the peaks as two independent variables. I wonder if the intensity of the peak changes as a result of changes in the chemical environment. If this is the case, some of the discussions on peak ratios later need to incorporate this dependency.

Figure 3: It is necessary to give more evidence to convince this reviewer that the 0.2eV memory effect is real, from the perspective of experimental uncertainty and peak fitting. Have they reproduced the results? Much of the discussion in that paragraph seems to be more on the results of NaCl(001) single crystals than of the particles. It is interesting to note the NaCl (001) data show another increase in shifts after reaching a plateau. Any reasons?

Line 190-210: The authors discussed that peak broadening of the particles is less than that of single crystals. I don't have a good sense of the discussion because there was no quantitative information for comparison. In Figure 1, compared to 0 mbar peak, the 5 mbar peak did occur to me that there was broadening. It would be good to again have some sensitivity analysis on how fitting and experimental uncertainty would potentially affect the extent of broadening or the lack of it. Are we talking about results that are statistically significant? Line 209-213: I cannot follow the logic. It was suggested earlier that "the decrease in peak broadening observed for single crystals is ultimately

attributed to adsorbed water reducing inhomogeneities in the surface potential". Do you mean that particles have more inhomogeneities and therefore the same amount of water will have a less effect in reducing inhomogeneities, when compared to single crystals? I am also confused by the term "immediately" in the sentence "The KPM experiments show that the inhomogeneities are removed immediately after water is adsorbed on the surface." What does it mean in terms of the amount of water needed to remove the inhomogeneities?

Line 217: How confident are the authors on these ratios, in light of the experimental uncertainty, especially that for Na 1s at high RH? Also, it is useful to include data of the UHV experiment and 0 mbar after dosing for comparison.

Line 268: The C-C/C-H peak dominates all the spectra of sucrose and may significantly affect the fitting of the C-O and O-C-O peaks in C1s. The authors need to do a better job in defending the validity of these fitted results.

Line 283: The O1s peak dominates in Figure 6 and the author attributed it to water vapor. I am surprised that whatever interactions between water vapor and the particles can lead to energy shift of the water vapor peak. Do you mean Adsorbed water? Did you see the same in NaCl experiments? Again, I am not sure how the uncertainty would affect the fitting results.

Line 292: I found the discussion " the changes in relative ratios between C-O and O-C-O bound carbon C 1s and oxygen O 1s signals are small" somewhat arbitrary. From eyeballing the peaks, I feel that there are some changes in the ratios of the intensities of the two peaks. If these changes are considered small, I would argue that the changes in Na/Cl ratios are small too.

Line 295: I found the use of EDB data to support their results not convincing, at least not consistent with the earlier claims that EDB, optical levitation and a host of other techniques do not provide adequate sensitivity in pre-deliquescence water uptake.

Line 340: Is the comparison of absolute intensities across different spectra (C1s and O1s) valid? I guess ok for observing a trend but may need more justification to compare with the stoichiometric ratio of malonic acid.

The authors may want to comment on the application of this technique at high RH values, which are more atmospherically relevant.

References: Cheng et al. (2013) Surface Chemical Composition of Size-fractionated Urban Walkway Aerosols Determined by X-ray Photoelectron Spectroscopy, Aerosol Science and Technology, 47, 1118-1124. DOI:10.1080/02786826.2013.824066. Gen M. and Chan C.K. (2017) Electrospray-surface enhanced Raman spectroscopy (ES-SERS) for probing surface chemical compositions of atmospherically relevant particles. Atmospheric Chemistry and Physics. https://doi.org/10.5194/acp-17-14025-2017. Gen M., Kunihisa R., Matsuki A., Chan C.K.* (2019) Electrospray surface-enhanced Raman spectroscopy (ES-SERS) for studying organic coatings of atmospheric aerosol particles. Aerosol Science and Technology. https://doi.org/10.1080/02786826.2019.1597964

---

## Referee Comment (RC2) · Anonymous Referee #2 · 3 Jul 2020

In this work, Lin et al. study the uptake of water vapor onto the surfaces of impacted NaCl, sucrose and malonic acid aerosols relative humidities from 0 to 16% using X-ray photoelectron spectroscopy, a surface-sensitive technique capable of detecting subtle chemical occurring with adsorption of water molecules. They find that water is adsorbed onto NaCl and sucrose particles at low RH, well below the RH at which particles deliquesce, but sucrose does not. Generally, I think this work is very solid and deserves prompt publication. I congratulate the authors on very thorough statistical analysis and literature analysis regarding the results. However, the specific importance of this measurement on aerosols is not emphasized. Similar measurements have been performed for non-aerosol samples of the same substances. This is the major shortcoming of the
work.

There are a few additional issues with the paper that frustrate the efforts of the reader to contextualize and cite the work. Furthermore, the fitting parameters, which are not included, should be more carefully documented and displayed somewhere in the paper or supplement. These improvements are necessary and could improve the impact and longevity of the work.

I recommend publication in ACP if these issues, and the following comments, can be addressed.

Major Comments

The literature review and discussion focus on aerosol water uptake, a phenomenon separate from water adsorption below deliquescence. More emphasis could be placed, in the introductory and concluding sections, on the chemistry occurring on surfaces of aerosol particles. Surface chemistry and reactions occurring during evaporation/condensation on aerosols is a separate and rapidly evolving branch of this science with many recent publications. This manuscript would have a larger impact if it included some references to recent developments in this topic. What reactions are promoted by adsorption of water molecules onto pre-deliquescent NaCl or carboxylic acids? This is an important question for this manuscript to discuss.

The conclusions are not prevalent enough. The assessment summarizing each qualitative/quantitative description are lost in each paragraph, although the results are presented with very good attention to detail and thoughtful analysis. I recommend improving the prevalence of these conclusion sentences, e.g. through their placement at the head of each paragraph, through an increased number of headings, or via another method. This would significantly improve the comfort of the reader and perhaps the breadth of the readership.

The peak fitting parameters are missing, although the fitting of peaks in this work is

described with great care. Further, displaying all the peak fits in the main text may not be necessary. The statistical analysis of these peaks (peak area, peak width, peak shifts) are more interesting. More emphasis on the statistics would be helpful in interpreting the quality of the data and the conclusions presented.

Minor comments

1. The fitted peaks in each figure are visually blocking the data points. Please rectify the situation.

2. In the introduction or XPS measurement section, a brief description of the measurement mechanism and e.g. the meaning of the signal, for a slightly broader audience, would be appreciated. This topic is of great interest to various readers who do not use XPS.

3. "2.3 Data Analysis" – the peak fitting is described carefully but it is not clear to me how much the peak shape is derived from first principles and how much is empirical. If the shape of these peaks is not physically meaningful, less emphasis could be placed on justifying the process of fitting. Where there is a meaningful connection between the equation and the data, this could be emphasized.

4. Line 163 – "after calibrating . . . as described earlier" – this sentence is not needed, especially at the top of the paragraph.

5. Lines 164-165 – parameters like signal-to-noise and error bars on the fits – the omission of which I feel are a major detriment to the paper – should be included in a table somewhere, or in the supplement.

6. Lines 192-195 and lines 209-211– How does drying the aerosol influence the crystal form significantly? This is one important way in which the aerosol measurement may prove different from the non-aerosolized measurements. In keeping with what I believe is the major shortcoming of this manuscript, this connection between your work and the aerosol in the atmosphere is important to discuss in a location and/or under a heading

where the reader can find it easily.

7. Lines 229-235 – the length of these statements could be reduced significantly.

8. Lines 242-245 – does curvature of the impacted particles resting on the substrate change the signal intensity corresponding to surface adsorption by virtue of the tilted angle of the sides of the particles? This is true of e.g. microscopy studies of impacted particles.

9. Lines 261-262 – specifically, how?

10. Figure 4 – the "COOH:COOH" looks very redundant here.

11. Lines 292-294, 301, and 303 – could these statements have come sooner in the section/paragraph?

12. Line 313 – please define "DP1" and "DP2"

13. Figure 8 – the peaks are very close together. It would be helpful to see a 95% confidence interval of the peak, or similar.

---

## Author Comment (AC1) · 13 Oct 2020

We thank the reviewers for their time and many insightful comments and suggestions. In response to comments from both reviewers, we have strengthened the connection of our measurements to processes involving real atmospheric aerosols throughout the paper. We thank reviewer # 1 for the suggestions of additional relevant literature concerning pre-deliquescent water uptake.

In response to both reviewers' concerns about the curve fitting process, we have redone the peaks fits including a thorough error analysis using a Monte Carlo method. This is done by replicating the measured spectrum as a noiseless version with a fit,

which itself has no physical meaning but repeats the overall structure of the measured spectrum. The Monte Carlo procedure then produces a number of virtual experiments by adding random noise to the noiseless spectrum and these virtual spectra are subsequently fitted under the same assumptions as the original measured spectra. The procedure is run 100-200 times, and the uncertainty of each fitting parameter is estimated as $\pm\, 2\sigma$, where $\sigma$ is the standard deviation of the set of parameters from the Monte Carlo simulations. A description of this error analysis has been added to the Data Analysis section 2.3. Results of the Monte Carlo error analysis are shown as error bars on experimental data and have also been included in the supplement for reference.

Below, we give a point-by-point response to each of the reviewers' comments and indicate corresponding changes made in the revised manuscript.

**Reviewer #1**

Specific comments:

1. Line 148: The authors discussed the shift in binding energy and the intensities of the peaks as two independent variables. I wonder if the intensity of the peak changes as a result of changes in the chemical environment. If this is the case, some of the discussions on peak ratios later need to incorporate this dependency.

   Binding energies of the core-level electrons depend on the chemical state or surroundings of the element in question. Changes in the chemical state or environment are directly reflected as shifts in the electron binding energies. In XPS experiments, the intensities–or more accurately, the areas–of the spectral peaks directly reflect the amounts of the element in question which is in the chemical states or environments contributing to each peak. A change in peak intensity or area reflects a loss of the element from the chemical states or environments contributing to that peak. For NaCl, where we use the full integrated elemental

peak areas to quantify the amounts of Na and Cl in the systems, a change in the respective peak areas would imply a loss of the element from the system. For the C 1s spectra of the organic aerosols, a change in chemical state of the carbon atoms would be reflected in equivalent opposing changes in the areas of the peaks corresponding to the states involved in the transformation. This proportionality is one of the major advantages of core-level XPS, compared to other surface sensitive techniques, where higher order effects may indeed affect the peak intensities in complex ways (e.g. Hüfner, 2003).

In this work, we use the total amount of Na and total amount of Cl (NaCl), total amount of C and total amount of O (sucrose), and amount of C and O in the carboxyl groups (malonic acid) for determining the peak area ratios. Changes in one of these ratios reflect changes in the amounts of each element from the chemical states used in obtaining the ratios, and therefore an effect of water on the chemical composition. Due to the relatively noisy data obtained in some conditions in our experiments, we cannot necessarily distinguish all the possible chemical states that could be formed in connection to the water adsorption onto the particle surfaces. From the present mesurements, we can therefore assert that a loss of peak intensity/area reflects a change, even though we cannot give more details on the chemical nature of this change.

In practice, processes such as charging effects from increasingly ionizing the sample during the XPS experiments can affect the apparent binding energy of the detected photoelectrons, leading to peak broadening. However, this does not change the overall the peak area.

We have tried to clarify these points in the revised manuscript.

2. Figure 3: It is necessary to give more evidence to convince this reviewer that the 0.2eV memory effect is real, from the perspective of experimental uncertainty and peak fitting. Have they reproduced the results? Much of the discussion in that paragraph seems to be more on the results of NaCl(001) single crystals than of

the particles. It is interesting to note the NaCl (001) data show another increase in shifts after reaching a plateau. Any reasons?

In this context, it is important to note that these type of experiments require very specialised systems (APXPS) which are not widely available. Most of the existing systems globally are hosted at synchrotron radiation facilities and accessed for a limited time period based on experiment time granted in a highly competitive application process. Therefore, we unfortunately did not have time to repeat all experiments during the time available for our experiments. Furthermore, data analysis is elaborate and key results may only become evident long after the experiment has been concluded. Nevertheless, each of the presented spectra are averages of tens of unique acquisitions (of entire spectra) and therefore in fact ensembles of individual measurements. We note that the individual spectral scans do not drift in energy with time, which means that the stability of the energy scale during the measurement period was very high. In addition, we performed uncertainty estimates for the spectral fits using the Monte Carlo procedure described above. We note that the uncertainties in the binding energy from the Monte Carlo simulations ($\pm\ 2\sigma$, confidence interval of 95%) are much smaller than the observed memory effect of 0.2 eV.

On a technical note, the energy step size used in the acquisition of our XPS spectra is 0.1 eV. This is greater than the energy accuracy (not to be confused with the resolution) of the analyser when operating at the 50 eV or 100 eV pass used in these experiments. We therefore consider an observed memory effect larger than 0.1 eV to be real and not explained by experimental uncertainties alone. A similar memory effect is also observed by Verdaguer et al. (2008) for the NaCl(001) crystal, which further supports our present findings for the aerosol samples.

These points have been clarified in the revised manuscript.

We agree that the discussion of our results relies strongly on the comparison

to previous results for NaCl(001). We consider this as key to show that our results are in agreement with previous work and thereby anchoring our experiments as feasible also for NaCl nanoparticle samples with a more complex physical structure and further on for the organic malonic acid and sucrose particle samples. The increase in electron binding energy shifts after the plateau observed by Verdaguer et al. (2008) has been attributed to more efficient discharging (dissipation of the charge buildup from sample ionization during the XPS measurements) of the NaCl surface due to enhanced mobility of surface ions by solvation into the adsorbed water layer. In the present experiments, we were not able to reach sufficiently high humidities to observe whether such an additional increase would be present also for the nanoparticle samples, but the Cl 2p binding energy shifts do indicate the presence of a plateau after the initial shift. As experimental facilities are continually developed and enabling us to reach higher relative humidities, we hope to have the possibility of studying this phenomenon further in the near future.

3. Line 190-210: The authors discussed that peak broadening of the particles is less than that of single crystals. I don't have a good sense of the discussion because there was no quantitative information for comparison. In Figure 1, compared to 0 mbar peak, the 5 mbar peak did occur to me that there was broadening. It would be good to again have some sensitivity analysis on how fitting and experimental uncertainty would potentially affect the extent of broadening or the lack of it. Are we talking about results that are statistically significant?.

We thank the reviewer for pointing this out. We here refer to changes in the peak broadenings, not to peak broadenings themselves. This has been clarified in the revised manuscript.

In our experiments with aerosol samples, the overall resolution is lower than that in the single crystal experiments of Verdaguer et al. (2008) and we do not observe significant changes in the widths of the peaks as a function of humidity.

We agree that the 5 mbar Na 1s peak does look broader than the corresponding peak at 0 mbar, but the statistics of the spectrum are poor. For the other humidity conditions, no significant changes in the peak broadening were observed. Therefore, the only meaningful information to report regarding the peak broadening in our experiments is the fact that it does not change between the investigated humidities.

We have performed Monte Carlo error analysis of our fits (see above) and included the resulting error estimation ($\pm\,2\sigma$) to give the uncertainties (confidence of approximately 95%) due to spectral fitting. We are therefore confident that the results are statistically significant and indeed show changes in the chemical and elemental composition of the particle surfaces.

4. Line 209-213: I cannot follow the logic. It was suggested earlier that "the decrease in peak broadening observed for single crystals is ultimately. attributed to adsorbed water reducing inhomogeneities in the surface potential". Do you mean that particles have more inhomogeneities and therefore the same amount of water will have a less effect in reducing inhomogeneities, when compared to single crystals?

Yes, this was what we meant. The decrease in the peak broadening for the single crystals is indeed attributed to reduced inhomogeneity due to adsorbed water. We attribute our observations for particle smaples to the fact that the nanoparticle surfaces are very different from single crystal surfaces and contain a lot of surface sites that are different from one another in terms of coordination and morphology, and therefore we do not necessarily expect the overall inhomogeneity to be reduced as efficiently as for single crystals. Especially the nanoparticles can contain morphologically sharp regions, where electric fields become larger. While the amount of chemical inhomogeneities may be reduced, we expect that morphological changes would require larger amounts of water, or dissolution, and therefore we do not expect that all electric fields or potentials will be removed from

the nanoparticle surfaces as efficiently as in the case of single crystals. We have clarified this point in the revised manuscript.

5. I am also confused by the term "immediately" in the sentence "The KPM experiments show that the inhomogeneities are removed immediately after water is adsorbed on the surface." What does it mean in terms of the amount of water needed to remove the inhomogeneities?

With "immediately" we meant to say that this happens already at very low water coverage and does not require the formation of a thick water layer on the particle surfaces. Our point is that the decrease in peak broadening occurs when only very few water molecules are adsorbed onto the crystal surface. This is clarified in the revised manuscript, avoiding the term "immediately", which indeed could unintendedly imply temporal aspects.

6. Line 217: How confident are the authors on these ratios, in light of the experimental uncertainty, especially that for Na 1s at high RH? Also, it is useful to include data of the UHV experiment and 0 mbar after dosing for comparison.

We have entirely redone the analysis related to the spectral fitting, with the same initial assumptions as in the original analysis, providing also a thorough error analysis for the peak area ratios. During this process, we spotted two systematic errors that were made in our original analysis when transferring data from the analysis software to the figures and manuscript. In the revised manuscript, we report the corrected values for the peak area ratios for NaCl and malonic acid and have made new figures with these corrected values, including error bars ($\pm\ 2\sigma$, 95% confidence interval) for the peak area ratios. We note that the overall trend of the malonic acid O:C ratio, as well as the Na:Cl ratio for NaCl, both remain unchanged with respect to our initially reported results.

We are confident that the changes in peak area ratios as shown in Fig. 3 are real and that they reflect changes in the chemical composition at the particle

surfaces. For malonic acid particles, we cannot at this time offer an explanation to the values of the observed O:C ratio or the behavior at different humidities. However, we chose to report our observations as potentially meaningful in the context of future experiments. For NaCl particles, the possible reasons for the changes in the Na:Cl ratio with humidity are discussed in the manuscript.

We have presented results for 0 mbar after dosing $H_2O$ (in the manuscript) and UHV (in the supplement) for NaCl particle samples. The Na:Cl ratio after dosing was determined as $0.62\pm0.11$ ($2\sigma$), or very nearly the same as before water dosing. This information has been added to the revised manuscript. For sucrose particles, similar results are not shown, as there were no significant changes observed between the conditions. For malonic acid, the "0 mbar" (inside the cell, without $H_2O$ vapor) before or after dosing were unfortunately not measured, due to measurement time constraints. The UHV data is essentially the same as the 0 mbar data, albeit with somewhat better statistics, and therefore does not give more information on the system.

7. Line 268: The C-C/C-H peak dominates all the spectra of sucrose and may significantly affect the fitting of the C-O and O-C-O peaks in C1s. The authors need to do a better job in defending the validity of these fitted results.

We are confident in the fitting of the C 1s C-O and O-C-O peaks. Literature values from the XPS spectrum of sucrose in vacuum (Stevens and Schroeder, 2009) give a sound initial assumption for the starting points of the spectral fits, and our binding energies for C-O and O-C-O at 0 mbar are in good agreement with those of Stevens and Schroeder (2009), as noted in line 274 of the original version of our manuscript. In addition, we have now redone the fitting analysis with 100 Monte Carlo simulations which yielded essentially the same results as before. Results of the re-analysis and Monte Carlo error estimation are presented in the updated Fig. 5.

[Figure]

8. Line 283: The O1s peak dominates in Figure 6 and the author attributed it to water vapor. I am surprised that whatever interactions between water vapor and the particles can lead to energy shift of the water vapor peak. Do you mean Adsorbed water? Did you see the same in NaCl experiments? Again, I am not sure how the uncertainty would affect the fitting results.

The strong peak in the O 1s spectra is indeed that of gas phase water. However, the binding energy of the water vapor does not change in our fits. The gas phase O 1s binding energies are 536.05±0.03, 536.06±0.02, and 536.03±0.01 eV for 0.2, 1, and 5 mbar $H_2O$, respectively. These numbers can now be found in the supplement. In line 280, we have clarified that it is the peak area of gas phase water, and therefore the total amount near the particle surfaces, that is increasing with relative humidity as expected.

We do observe a shift in binding energy of the individual O-C-O and C-O O 1s peaks relative to the gas phase water peak–about 0.5 eV going from 0 mbar to 5 mbar. This could be due to the influence of adsorbed water, but we cannot say for certain since the binding energy shift in both the O 1s and C 1s peaks are small. The shift could also be explained by change in the work function of the particle surfaces due to the presence of gas phase molecules (Axnanda et al., 2013). This appears as an apparent binding energy shift that would have a proper reference in ultra-high vacuum.

For the NaCl particle samples, we did not show any O 1s spectra, because we were unable to resolve the adsorbed water signal due to the strong signal from both gas-phase water and silicon oxide from the substrate. The malonic acid and sucrose particles were deposited on gold foil and therefore do not have the same issue with substrate oxide signal.

9. Line 292: I found the discussion " the changes in relative ratios between C-O and O- C-O bound carbon C 1s and oxygen O 1s signals are small" somewhat arbitrary. From eyeballing the peaks, I feel that there are some changes in the

ratios of the intensities of the two peaks. If these changes are considered small, I would argue that the changes in Na/Cl ratios are small too.

We agree with the reviewer that the changes are subtle, however they are clear, for both sucrose and NaCl particles. We have presented the corresponding error bars from the Monte Carlo simulations and clarified the discussion on this point in the revised manuscript.

10. Line 295: I found the use of EDB data to support their results not convincing, at least not consistent with the earlier claims that EDB, optical levitation and a host of other techniques do not provide adequate sensitivity in pre-deliquescence water uptake.

We agree that the comparison to EDB experiments could seem inconsistent, given that we present the results from XPS measurements as providing information which is generally not accessible with bulk-sensitive methods. The reference to EDB experiments was made, because these measurements provide some of the only data that our present results can be immediately compared against. We have reformulated this section to clarify that our XPS measurements with their higher sensitivity to condensed water confirm observations from previous EDB experiments.

11. Line 340: Is the comparison of absolute intensities across different spectra (C1s and O1s) valid? I guess ok for observing a trend but may need more justification to compare with the stoichiometric ratio of malonic acid.

No direct comparison is made between the absolute peak areas for C or O, only the C 1s to O 1s peak area ratio. In the data analysis we have taken the different measurement effects which are affecting the peak area ratios into account, by normalizing spectral peak areas to the photoionization cross section of each orbital and the attenuation from scattering of the photoelectrons in the water vapor.

The transmission function of the analyzer is constant at these kinetic energies (SPECS Surface Nano Analysis GmbH).

12. The authors may want to comment on the application of this technique at high RH values, which are more atmospherically relevant.

   Our presented experiments cover relative humidities up to 16%, however, with further optimization of the experimental system, it will be possible to cover the full range of relative humidities from 0 to 100%, further enabled by combination with ultra-high brilliance synchrotron X-rays. We have expanded the discussion in the conclusions section about the applicability of APXPS at higher RH values.

**Reviewer #2**

The reviewer makes the important point, that "...the specific importance of this measurement on aerosols is not emphasized. Similar measurements have been performed for non-aerosol samples of the same substances. This is the major shortcoming of the work."

We agree that this is a very important aspect of our presented work, which we have now tried to highlight more clearly in the revised manuscript. We thank the reviewer for their reflection of this and related aspects of implications for atmospheric aerosol processes, which are indeed at the heart of our fundamental motivation for this work. XPS is a powerful and well-known technique in surface science which has recently been applied to systems of more immediate resemblance to atmospheric aerosols. Combined with ambient pressure conditions, the possibilities emerge for also studying processes of immediate atmospheric relevance, including adsorption and desorption from surfaces, and heterogeneous and surface-specific chemistry.

The main aim of our present work has been to investigate the potential for obtaining meaningful results with APXPS for aerosol samples comprising atmospherically relevant chemical components. Considering the additional degrees of freedom in terms of

variations in size, morphology and sample coverage introduced by the aerosol samples, this was not given a priori to be feasible. We have therefore used simple aerosol compositions and focused on contrasting our results to previous APXPS experiments on relatively simpler, macroscopic single crystal samples. Obtaining results which are in line with these previous measurements serves to benchmark the APXPS measurements for NaCl aerosol samples and allows us to have confidence in the results also for the aerosol samples comprising major atmospheric organic functional groups. The ability to obtain meaningful results with APXPS from deposited aerosol particles of atmospheric relevance opens for the application of this method to a wide range of laboratory-based studies of surface processes. This potentially enables direct and highly surface sensitive investigations of numerous atmospheric relevant processes with very high chemical selectivity.

Our present results confirm that XPS can be used to specifically observe chemical changes on aerosol surfaces which likely cannot be resolved with even high-resolution bulk methods, such as aerosol mass spectrometry. Several recent studies have shown unique features of surfaces, compared to the bulk phase, for systems of atmospherically relevant composition (e.g. Werner et al., 2018, and references therein). We therefore anticipate that the ability to directly probe aerosol surfaces with high sensitivity and chemical selectivity will enable new studies of surface specific processes and chemical properties with atmospheric significance.

Specifically related to the uptake of water to the aerosol phase, we anticipate that the presence of water at the aerosol surface could impact water catalyzed chemical reactions, as well as reactions involving mobilization of charged species such as organic and inorganic ions or components with acid-base functionalities. The existence of an aqueous layer may also impact adsorptive gas–particle equilibrium of other semi-volatile species in addition to water and in turn their further chemical fate in the atmosphere. The main significance of our current findings is that these processes could occur as a consequence of water adsorption in even very dry conditions. The

surface–to–bulk volume ratios of finite-sized atmospheric aerosols are orders of magnitude larger than for macroscopic systems (Prisle et al., 2010b; Bzdek et al., 2020) and consequently even processes which are confined to the immediate surface region could significantly impact the overall aerosol transformation.

We have highlighted these aspects in the revised manuscript. Speculating on the exact nature and magnitude of the vast number of possible chemical and physical transformations of atmospheric aerosol surfaces is however outside the scope of this work.

Other major comments:

1. The literature review and discussion focus on aerosol water uptake, a phenomenon separate from water adsorption below deliquescence. More emphasis could be placed, in the introductory and concluding sections, on the chemistry occurring on surfaces of aerosol particles. Surface chemistry and reactions occurring during evaporation/condensation on aerosols is a separate and rapidly evolving branch of this science with many recent publications. This manuscript would have a larger impact if it included some references to recent developments in this topic. What reactions are promoted by adsorption of water molecules onto pre-deliquescent NaCl or carboxylic acids? This is an important question for this manuscript to discuss.

   We have focused the literature review and discussion on observations of aerosol water uptake and pre-deliquescent water uptake to salt surfaces, since these provide the immediate context of our present measurements. The direct observation of water adsorption to surfaces of aerosol particles at very low humidities could indeed have significant implications for aerosol surface and heterogeneous chemistry, for example via mechanisms as suggested above. It could furthermore readily be speculated that other volatile or semi-volatile atmospheric components similarly adsorb at aerosol surfaces, in quantities which go unnoticed in bulk-sensitive measurements, but with ability to significantly alter the chemical state and further chemical transformation of molecules in the top-most surface layers of the aerosol.

The reviewer makes an excellent point, so in the revised manuscript we have added discussion of recent aerosol surface chemistry and reactions to the introduction as context for our measurements.

The surfaces of aerosol and droplet particles are distinct physical and chemical environments compared to their associated bulk phases. The composition of the droplet surface can influence the mass transport and chemical reactions that occur at the surface (e.g. Cosman et al., 2008; Park et al., 2009; Roy et al., 2020). The acidity of organic acids on water surfaces has been measured to be much lower than predicted for the bulk aqueous phase (Enami et al., 2010; Werner et al., 2018). The presence of surface-active organic molecules on droplet surfaces can affect droplet surface tension (Shulman et al., 1996; Bzdek et al., 2020) and morphology (Kwamena et al., 2010) that affect both warm (Sareen et al., 2013; Ovadnevaite et al., 2017) and ice cloud nucleation (Knopf and Forrester, 2011; Perkins et al., 2020). The formation of an aqueous phase can lead to the partitioning of water-soluble gases to the condensed phase (Prisle et al., 2010a), including many reactive oxidants (Donaldson and Valsaraj, 2010), that can initiate a wide range of aqueous phase chemistry (McNeill, 2015).

A number of aqueous phase reactions occur between inorganic salt species and organic compounds. The hygroscopic properties of sodium halide particles coated with fatty acids depend on both the salt anion and the carboxylic acid, with some mixtures showing barriers to water uptake while others do not (Miñambres et al., 2014). Depletion of chloride and bromide from marine aerosol particles increases under the influence of biogenic wildfire emissions that contribute organic acid to the aerosol (Braun et al., 2017). Enhanced production of sulfate aerosol via nitrate photolysis was observed to be facilitated by the presence of surface-active halide ions (Zhang et al., 2020).

2. The conclusions are not prevalent enough. The assessment summarizing each qualitative/quantitative description are lost in each paragraph, although the results are presented with very good attention to detail and thoughtful analysis. I recommend improving the prevalence of these conclusion sentences, e.g. through their placement at the head of each paragraph, through an increased number of headings, or via another method. This would significantly improve the comfort of the reader and perhaps the breadth of the readership.

   We thank the reviewer for this observation and have done our best to restructure the discussion to emphasize the main conclusions throughout the revised manuscript.

3. The peak fitting parameters are missing, although the fitting of peaks in this work is described with great care. Further, displaying all the peak fits in the main text may not be necessary. The statistical analysis of these peaks (peak area, peak width, peak shifts) are more interesting. More emphasis on the statistics would be helpful in interpreting the quality of the data and the conclusions presented.

   We now present the peak fitting parameters, specifically binding energy, peak area, Lorenzian FWHM, Gaussian FWHM, total FWHM, and asymmetry parameter in the tables in section S1 of the supplement. We have clarified the description of these parameters and their relation to the analysis in the revised manuscript. The main parameters used in our analysis–photoelectron binding energy, peak area, and total FWHM–are described in the main text. The remaining peak fitting parameters are described in the supplement for reference and completeness. Following the Monte Carlo error estimation of the peak fitting parameters, we have strengthened the emphasis on statistics in the data analysis.

Minor comments:

1. The fitted peaks in each figure are visually blocking the data points. Please rectify the situation.

   We thank the reviewer for pointing this out. We have made sure the data points are plotted on top of the fits in Figs. 1–2, and 5–8.

2. In the introduction or XPS measurement section, a brief description of the measurement mechanism and e.g. the meaning of the signal, for a slightly broader audience, would be appreciated. This topic is of great interest to various readers who do not use XPS.

   We thank the reviewer for this very useful suggestion. We have clarified the key concept of electron binding energy underpinning the XPS measurements and the resulting XPS signal in the general experimental section and the significance of the electron binding energies and peak areas determined from the XPS spectra in the data analysis section. Specifically, we have made the following changes:

   l. 70-75: *"Photoelectron spectroscopy utilizes the photoelectric effect, by which the sample is ionized from inelastic collisions with photons and the emitted electrons are detected and characterized in terms of their kinetic energy ($E_k$). When the ionizing photon energy ($h\nu$) is known, the binding energy ($E_b$) of electrons within the sample can be determined simply as $E_b = h\nu - E_k$. By using X-ray photons, core-level atomic-like orbitals can be ionized, and the electron binding energy gives a very sensitive fingerprint of the chemical composition and environment of the sample. XPS is furthermore a highly surface-sensitive technique, because the resulting kinetic energies of the photoelectrons yield very short characteristic attenuation lengths and the detected photoelectron signal originates mainly from the topmost few nanometers of the sample. An XPS measurement consists of measuring the intensity of photoelectrons emitted from the sample as*

*a function of the electron kinetic energy. Typically, an XPS spectrum presents the photoelectron signal intensity as function of the orbital binding energy and consists of a collection of peaks, each corresponding to a different chemical species or environment. Here, we quantify the spectral peaks in terms of their areas, which are directly proportional to the relative abundances of each species on the surface of the sample. Spectral fitting techniques are employed to obtain accurate results for both binding energies and peak areas.*

The second paragraph of section 2.3: *"The aim of the spectral fitting procedure is to characterize the measured spectra in terms of peak position and intensity. The position of a given peak gives the binding energy of the core electron for a given element. Changes in the binding energy as well as the width of the fitted peaks–or peak broadening–indicate changes in the chemical environment or physical state of the sampled surface. The area of the peak is directly proportional to the amount of the element being measured. For the analysis here, we determine the elemental composition of particle surfaces as the relative ratios of the core level peak areas. The peak area of the XPS signal depends on a number of factors, including experimental parameters of the incident radiation and electron spectrometer as well as physical and environmental properties affecting the orbital from which the photoelectron originated. If all of these parameters are known, the XPS signal can be used to quantify the amount of species $i$. While these parameters are not always known, comparison of XPS signals is still possible to quantify differences in elemental abundances and chemical states between experimental conditions. Before extracting relative ratios of the peaks, all spectra were normalized to the photoionization cross section (Yeh and Lindau, 1985) of the given core electron. The attenuation of photoelectron intensity due to scattering of the photoelectrons from the water vapor was estimated by using the kinetic theory formulation (Ogletree et al., 2009) and measured electron scattering cross section data (Muñoz et al., 2007). The attenuation must be taken into account, because the fixed excitation energy from the X-ray source leads to significantly*

*different kinetic energies of the emitted photoelectrons and consequently different mean free paths in the vapor environment."*

3. "2.3 Data Analysis" – the peak fitting is described carefully but it is not clear to me how much the peak shape is derived from first principles and how much is empirical. If the shape of these peaks is not physically meaningful, less emphasis could be placed on justifying the process of fitting. Where there is a meaningful connection between the equation and the data, this could be emphasized.

Yes, the peak shape is derived from first principles. Photoelectrons experience a lifetime broadening effect from the uncertainty principle due to the lifetime of the core-hole created by an incident photon. This broadening is represented by a Lorentzian shape. The peak shape is also affected by measurement uncertainties that are best represented with a Gaussian shape. Together, these two broadening effects are represented with a Voigt function, or the convolution of a Lorentzian and a Gaussian (Jain et al., 2018).

We have clarified these aspects in the revised manuscript and moved some of the more technical discussion of the peak fitting process to section S1 of the supplement.

4. Line 163 – "after calibrating . . . as described earlier" – this sentence is not needed, especially at the top of the paragraph

We have removed "as described earlier" from the sentence.

5. Lines 164-165 – parameters like signal-to-noise and error bars on the fits – the omission of which I feel are a major detriment to the paper – should be included in a table somewhere, or in the supplement.

We give uncertainty estimates from the Monte Carlo analysis for the binding energy, peak area, Lorenzian FWHM, Gaussian FWHM, total FWHM, and asymmetry parameter in the tables in section S1 of the supplement. These parameters

give quantification of the quality of the fits. We do not consider that calculating the exact signal-to-noise ratios will provide more information on the fit quality than what can be immediately assessed from the spectra shown in Figs. 1–2 and 5–8.

6. Lines 192-195 and lines 209-211– How does drying the aerosol influence the crystal form significantly? This is one important way in which the aerosol measurement may prove different from the non-aerosolized measurements. In keeping with what I believe is the major shortcoming of this manuscript, this connection between your work and the aerosol in the atmosphere is important to discuss in a location and/or under a heading where the reader can find it easily.

Results of our study show that the aerosol particle samples investigated have more complex morphology than the simple single crystal surfaces previously studied by XPS. Several previous studies have observed that the process of drying an aerosol can indeed affect its crystalline form. For example, studies of NaCl aerosol particles generated from drying of aqueous droplets have inferred a non-crystalline structure with pores or pockets that trap liquid water (Weis and Ewing, 1999; Darr et al., 2014; Braun and Krieger, 2001). This is explained by the presence of liquid water detected below the deliquescence relative humidity but at much higher RH than in our study. Furthermore, the morphology of NaCl particles expressed via the shape factor has been shown to be controlled by the drying rate (Wang et al., 2010). A recent study (Archer et al., 2020) has explained the morphology of particles formed from drying of a colloid as a competition between diffusion of solute in solution versus loss of solvent, with higher solvent loss rate compared to solute diffusivity leading to more complex morphologies. For atmospheric samples, microscopy studies on sea salt particles has shown them to have complex morphologies (e.g. Cheng et al., 1988), similarly to what was found for the laboratory generated aerosol samples in the present study.

Atmospheric aerosols are likely to undergo drying and humidification cycles under a wide range of conditions and thus to exhibit a range of morphologies re-

ACPD

lated to drying. Our measurements on aerosol particle samples generated from nebulization and subsequent dessication are therefore expected to much more closely represent the morphologies of atmopsheric aerosols, compared to the simple uniform morphologies of single-crystal samples.

We have added this discussion to Section 3.1 in the manuscript.

7. Lines 229-235 – the length of these statements could be reduced significantly.

We have tried to simplify and clarify this section. It was not possible to reduce the length significantly at the same time.

"To quantify the attenuation of the photoelectron signal, we use an exponential decay function $I_n = I_n^0 e^{-x/\lambda_n}$, where $I_n$ is the attenuated intensity of peak $n$, $I_n^0$ is the corresponding unattenuated intensity, $x$ is depth into the sample from where the signal originates, and $\lambda_n$ the energy-dependent inelastic mean free path of photoelectrons contributing to peak $n$. The depth of origin can be expressed as $x = \frac{\lambda_1 \lambda_2}{\lambda_1 - \lambda_2} \ln R$, where $R = \frac{I_1 I_2^0}{I_2 I_1^0}$ is the relative ratio of attenuated and unattenuated signals from two separate peaks $n = 1, 2$. We here use the total integrated peak areas to represent signal intensities. In our experiments, the unattenuated signal ratio (measured without water vapor) for Na 1s and Cl 2p is $I_{\text{Na}}^0 / I_{\text{Cl}}^0 = 0.9$. With this, from the corresponding signal ratios at elevated humidities, the simple attenuation model gives depths of photoelectron origin (or water layer thickness) of approximately 14 and 4 Å for 6.3 and 16% RH, respectively. This is much more than the previous observations of 2.4 Å by Cabrera-Sanfelix et al. (2007) and also counterintuitive as it would mean decreasing layer thickness with increasing RH."

8. Lines 242-245 – does curvature of the impacted particles resting on the substrate change the signal intensity corresponding to surface adsorption by virtue of the tilted angle of the sides of the particles? This is true of e.g. microscopy studies of impacted particles.

This is an interesting question. Given that we are here studying an ensemble of submicron particles using an X-ray beam with a 10 mm spot size, we do not expect to see an effect from the curvature of individual particles on the measured photoelectron signal intensities in these experiments. Since our analysis is based on peak area ratios, any effect would not impact the conclusions of this work.

9. Lines 261-262 – specifically, how?

We agree this paragraph was poorly formulated. We have rephrased lines 258–62 in the original manuscript to:

"Using XPS on free-flying sub-2 nm CsBr water clusters, Hautala et al. (2017) found that Br$^-$ ions were closer to the surface that their counter cations. This supports the interpretation of our present observations that pre-deliquescent water adsorption enhances chloride relative to sodium in the aerosol surface layer. The presence of halide ions, especially Cl$^-$ and Br$^-$, at the air-water interface has been connected to increased photochemical activity (e.g. George and Abbatt, 2010, and references therein). The mobilization of ions can lead to release of gaseous halogen compounds from sea salt aerosol due to reactions in the aqueous phase (Mozurkewich, 1995; Vogt et al., 1996; Kerminen et al., 1998; Keene et al., 1999). In the atmosphere, formation of solvated halogen ions even at very dry conditions via similar pre-deliquescent adsorption of water onto the surfaces of sea salt aerosol as seen for laboratory generated aerosol in this work could therefore have significant implications for the halogen cycle, including ozone chemistry in the polar regions (Simpson et al., 2007)."

10. Figure 4 – the "COOH:COOH" looks very redundant here.

We are not entirely sure what is meant here. The notation C**OO**H:**C**OOH where the oxygen and carbon atoms, respectively, are highlighted in boldface, refers to the ratios of the O 1s to C 1s peak areas from the carboxyl groups of malonic acid. We agree that the boldface emphasis may unfortunately not be easy to

read, however, we have tried to clarify this point in the caption of Fig. 4 and in the main text of the revised manuscript. We have also taken the opportunity to streamline the legend and trace labels in the figure.

11. Lines 292-294, 301, and 303 – could these statements have come sooner in the section/paragraph?

We have rewritten these paragraphs in accordance with the reviewer's second major comment.

12. Line 313 – please define "DP1" and "DP2"

We have clarified that these are the decomposition peaks observed by Ferreira Jr. et al. (2017).

"The decomposition peaks DP1 and DP2 observed by Ferreira Jr. et al. (2017) were included in the fit for 0% RH (UHV), where a good fit could not be produced without their inclusion."

13. Figure 8 – the peaks are very close together. It would be helpful to see a 95% confidence interval of the peak, or similar.

The binding energy uncertainties for each peak are now included in Tables S1–S8 in the supplement.

[revised manuscript text omitted]

---

## Author Response (AR2)

**Authors' response to editor's comments for "Pre-deliquescent water uptake in deposited nanoparticles observed with *in situ* ambient pressure X-ray photoelectron spectroscopy"**

Jack J. Lin, Kamal Raj R, Stella Wang, Esko Kokkonen
Mikko-Heikki Mikkelä, Samuli Urpelainen and Nønne L. Prisle

January 11, 2021

We thank the editor, Thorsten Bartels-Rausch, for his time and careful reading of the manuscript and helpful suggestions. We address his comments below.

"My main remaining comment also addresses the uncertainty. May I ask you to elaborate in slightly more detail on the BE shifts that you observe and on the peak-ratio with water vap[o]r and rule out the following hypothesis. Please don't get me wrong, I think this is a very clever and novel analysis with a clear finding. However, the I believe the manuscript would be stronger if the following arguments are tackled explicitly:"

1. What is the origin of the shift you observe? I must confess, that I find the presentation of the results slightly confusing. I tend to understand that "Shift" is defined as change in observed or apparent binding energy relative to that at 0 mbar in the manuscript While this is correct; I'm used to a definition of "shift" relative to un-charged samples (or literature values). NaCl appears to have a N[a]1s BE of 1071-1072 eV; I would therefore argue that the 0mbar samples in Figure 1 are shifted, but the 2 and 5 mbar are not (or less). So, there is less (or no) shift with increasing water vapor pressure – as opposed to my understanding of lines 199-201. As your write, this shift is caused by charging of the sample in UHV and is reduced in presence of gas-phase water. The importance of this, I think, becomes evident, when looking at Figure 5 which describes the C1s of Sucrose. The C-H is precisely where one would expect it at 285eV. Apparently this sample is not charged even at 0 mbar. Then, is my point, this would explain why the BE does not change in presence of water as the sample has already been dis-charged at 0 mbar. This brings me to some questions that I ask you to discuss in your manuscript.

   Since we are primarily interested in changes in our samples with respect to the addition of water vapor, we do use "shift" to mean relative to the

measurements at 0 mbar. We understand that this may be confusing to those used to thinking of a shift in different terms. We have clarified what we mean by shift in the manuscript.

At 0 mbar, the adventitious carbon C-H peak is at 285.90 and 285.75 eV for sucrose and malonic acid, respectively, so they are not exactly at 285 eV. The binding energies from all the peak fits may be found in the supplement. For sucrose, the shift in binding energy is less than 0.5 eV up to 5 mbar water vapor pressure, which leads us to conclude that there is very little water uptake to the sucrose particles. For malonic acid, there is a shift of just over 0.5 eV at 1 mbar water vapor pressure. Along with other changes in the malonic acid C 1s and O 1s spectra with RH, we conclude that there is some water adsorption onto the malonic acid particles.

(a) Can you quantify how much dis-charging you would have expected by gas-phase water alone as compared to adsorbed water? That is an interesting finding, does this explain why and result in water gas being so much more efficient in discharging than nitrogen gas or even oxygen gas.

In their measurements of NaCl charge state, Verdaguer et al. (2008) controlled the relative humidity by keeping the water vapor pressure constant and changing the ambient temperature. Since the water vapor pressure was constant throughout the measurements, changes in the NaCl charge state were attributed to the adsorption of water. Since we see similar behavior as a function of RH to Verdaguer et al. (2008), we expect adsorption to similarly play a larger role in our measurements. However, since we control our relative humidity by changing the water vapor pressure, it may not be possible to exactly quantify the relative contribution to discharging from gas-phase water compared to adsorbed water from this set of data.

(b) How well are the samples electrically conductive? I assume that gold is very conductive, but have no experience with silicon wafers. Could it be that those are less conducting and therefore you observe a charge with the NaCl samples at 0 mbar and not with the organic samples on gold?

We do not have any quantitative numbers, but yes, we do expect the silicon wafers to be less conductive than gold. This could lead to different amounts of charging between the substrates, but we must stress again that we examine shifts in BE for a given compound on a given substrate relative to the BE at 0 mbar water vapor pressure.

(c) How did your sample look like? Could it be that you sample adventitious carbon mostly at the gold surface -without any charging even at 0 mbar - and sucrose as deposits with are charged at 0 mbar and with reduced charging at 1 mbar? I think this point might be stressed more in the manuscript to explain the different behavior of

the individual features in the C1s spectra with respect to changes of the BE at varying RH.

We are not entirely sure what the editor means here. Visually speaking, the gold substrates with organic compounds deposited onto them are indistinguishable from clean substrates to the naked eye. It could very well be that adventitious carbon is mostly at the gold surface and does not experience charging while the sucrose particles are on top of the adventitious carbon and do experience charging at 0 mbar water vapor pressure. However, we cannot rule out additional adventitious carbon accumulation after the deposition of the sucrose particles. We do not have sufficient sensitivity in our current measurements to comment with any certainty about the layering of the deposited particles and adventitious carbon.

(d) How certain are you about the BE of the C1s features. What is the impact on fitting (width, peak ratio, position contains as well as background treatment) on the BE for each feature and its change with RH?

We are confident in the BE of the C 1s features. We have used literature values from XPS measurements in vacuum conditions– Ferreira Jr. et al. (2017a,b) for malonic acid and Stevens and Schroeder (2009) for sucrose–to initialize our fits and subsequent Monte Carlo error analysis shows small uncertainty estimates in the binding energy.

(e) Adventitious carbon often also has some oxidized functionalities (Barr, 1998). Would one not need to differentiate more clearly between the CO peaks of your sample and those from the adventitious carbon (line 350). I agree that this gives too much freedom to the fits though.

In our fitting of the C 1s spectra of malonic acid and sucrose, we have followed Ferreira Jr. et al. (2017b) and Stevens and Schroeder (2009), respectively, in only assigning C-C/C-H bound carbon as adventitious carbon. It may be possible to account for oxidized functionalities of adventitious carbon in our measurements, but we agree that the overlap with the C-O peaks from the sample would probably give too much freedom in the fits.

2. Was there no adventitious carbon on the NaCl samples? By the way, why is O1s not shown for the NaCl samples to quantify the water uptake? Might masking by carbon also explain the trend seen in Fig. 4 and line 286-290 – assuming that carbon built up with time or that the data are from different sample spots?

This is an excellent point about adventitious carbon on the NaCl samples. Unfortunately, we did not check for C 1s on the NaCl samples, although it is almost certainly present. We do not believe that time or position-dependent masking by carbon explains the trend seen in Fig. 4 and lines

286-290. We do not expect any build up of adventitious carbon on the sample once the sample is in the measurement chamber, and we have kept the sample position fixed once we have found a suitable position for measurement.

As for O 1s for the NaCl samples, the contribution from the native oxide of the silicon wafer dominates the signal preventing any quantification of water uptake using the O 1s spectra. This is discussed in lines 209-212 of the manuscript.

Further comments (that you may or may not consider). I add a few references from our work because I think they might interest you. As editor it is not my intention to make you cite them.

1. Line 35: I could not agree more on the importance of the surface to volume ratio (Artiglia, Nat. Commun. 2017)

2. Line 50: Again, very important point. The presence of organics indeed may have profound impacts on heterogen[e]ous chemistry, in particular the solubility of ozone in the liquid aerosol phase and thus the reaction rate of halogen oxidation by gas-phase ozone (Edebeli, Env. Sci. Process Impacts 2019).

   We are glad that you agree with the importance of the two preceding points, and thank you for bringing the two papers to our attention.

3. Line 65: Here, I wonder how much aerosol deposits as you elegantly used differ from experiments with slurry considering that one exposed the samples to UHV and then humidifies again. Probably a topic to discuss during a conference – I hope we have the chance soon. (Lampimäki, J. Phys. Chem. 205; Orlando J Phys Chem Lett, 2019; Orlando Top. Catalysis 2016)

   Unfortunately we do not have any experience with slurries to make a comparison, but we would indeed be interested to discuss this further in the future.

4. Line 95: sampling nano-scale particles I have a practical question: How did you find the sample for a good XPS signal? Was it covering the whole sample holder at the measurement spot?

   Coverage on the sample substrate was sufficient that finding the sample was generally not a problem. We tried to optimize the position of the sample in order to maximize counts seen in Na 1s for NaCl particles on O 1s for the organic compounds.

5. Line 139: please define ESP

   The nanometer aerosol sampler (NAS) introduced in line 136 is a type of electrostatic precipitator (ESP). We now refer to NAS instead of ESP in line 139.

6. Table 1: Please define mü, sigma, etc in the caption.

The caption now reads: "Aerosol sampling data including generated size distribution information (geometric mean number $\mu_N$, geometric number standard deviation $\sigma_g$, total number $N$, and geometric mean surface area $\mu_{SA}$) and sample collection parameters (sampler flow rate $Q$, collection time $t$, substrate, and coverage)."

7. Line 200 ff: Water uptake before deliquescence has been observed before. Please cite Wise 2008, Aerosol Sci Technol 42, 281-294

We have added the findings of Wise et al. (2008) to the introduction in the paragraph beginning with "A number of spectroscopic techniques ..."

8. Figure 1: mention and explain blue peak in caption.

The caption now reads: "Na 1s XPS spectra of NaCl aerosol particles recorded at different water vapor pressures (relative humidities, RH). The point markers show the recorded data and the solid lines the fit envelope curve. The spectra are fit using a single, symmetric Voigt peak shown in red with an additional peak in blue necessary to explain the spectrum at 0 mbar after water exposure. The dashed vertical line shows the binding energy of Na 1s at 0 mbar pressure (0 % RH) at the beginning of the experiments. Error bars show the estimated uncertainty in the peak position from Monte Carlo analysis. Photon energy was 1486.6 eV from the Al anode."

9. Line 293 "This behavior" are you referring to the ratio at 0 mbar or the shift with RH?

We are referring to the ratio of Na to Cl calculated from the peak area ratios. We have clarified the sentence: "As the spectra for all RH are recorded with a constant excitation energy from the Al K$\alpha$ anode, different transmission through the electron analyzer at different electron kinetic energies cannot explain the difference between the Na to Cl peak area ratio and the NaCl stoichiometric ratio."

10. Line 310: Maybe it is worth to compare the surface coverage also to the findings by Ewing 2005 (H2O on NaCl: From single molecule, to cluster, to monolayer, to thin film, to deliquesence. In chapter 12, springer-Verlag, 2005

We cite Peters et al. (1997); Peters and Ewing (1997); Foster and Ewing (2000) for the surface coverage of water on NaCl (100), which are the same sources cited in Ewing (2005).

11. Line 338 that → than (?)

Yes. This has been corrected.

12. Line 358. Do you observe a difference in peak wi[d]th for gas-phase and condensed phase peaks? If so, I would add this information as strong argument.

For sucrose, the peak widths for gas-phase and condensed-phase water are different. However, for malonic acid, both gas-phase and condensed-phase water were assumed to have the same width. We have made these decisions in order to obtain the best fit to the measured data given the assumptions were could make to constrain the fit.

**References**

Ewing, G. E.: $H_2O$ on NaCl: From Single Molecule, to Clusters, to Monolayer, to Thin Film, to Deliquescence, in: Intermolecular Forces and Clusters II, pp. 1–25, Springer-Verlag, Berlin/Heidelberg, 2005.

Ferreira Jr., J. M., Trindade, G. F., Tshulu, R., Watts, J. F., and Baker, M. A.: Introduction to a series of dicarboxylic acids analyzed by x-ray photoelectron spectroscopy, Surface Science Spectra, 24, 011 001–5, 2017a.

Ferreira Jr., J. M., Trindade, G. F., Tshulu, R., Watts, J. F., and Baker, M. A.: Dicarboxylic acids analysed by x-ray photoelectron spectroscopy, Part I - propanedioic acid anhydrous, Surface Science Spectra, 24, 011 101–8, 2017b.

Foster, M. C. and Ewing, G. E.: Adsorption of water on the NaCl(001) surface. II. An infrared study at ambient temperatures, The Journal of Chemical Physics, 112, 6817–6826, 2000.

Peters, S. J. and Ewing, G. E.: Water on Salt: An Infrared Study of Adsorbed $H_2O$ on NaCl(100) under Ambient Conditions, The Journal of Physical Chemistry B, 101, 10 880–10 886, 1997.

Peters, S. J., Langmuir, G. E., and 1997: Thin film water on NaCl (100) under ambient conditions: An infrared study, ACS Publications, 13, 6345–6348, 1997.

Stevens, J. S. and Schroeder, S. L. M.: Quantitative analysis of saccharides by X-ray photoelectron spectroscopy, Surface and Interface Analysis, 41, 453–462, 2009.

Verdaguer, A., Segura, J. J., Fraxedas, J., Bluhm, H., and Salmeron, M.: Correlation between Charge State of Insulating NaCl Surfaces and Ionic Mobility Induced by Water Adsorption: A Combined Ambient Pressure X-ray Photoelectron Spectroscopy and Scanning Force Microscopy Study, The Journal of Physical Chemistry C, 112, 16 898–16 901, 2008.

Wise, M. E., Martin, S. T., Russell, L. M., and Buseck, P. R.: Water Uptake by NaCl Particles Prior to Deliquescence and the Phase Rule, Aerosol Science and Technology, 42, 281–294, 2008.

---

## Author Response (AR3)

**Authors' response to editor's comments for "Pre-deliquescent water uptake in deposited nanoparticles observed with *in situ* ambient pressure X-ray photoelectron spectroscopy"**

Jack J. Lin, Kamal Raj R, Stella Wang, Esko Kokkonen
Mikko-Heikki Mikkelä, Samuli Urpelainen and Nønne L. Prisle

February 1, 2021

We thank the editor, Thorsten Bartels-Rausch, for his dedication to improving this paper. We address his comments below. In addition, we have updated some of the binding energy values that were not updated in the last revision of the manuscript after the new Monte Carlo analysis of the fits. The results of the analysis remain unchanged. We have also made some minor edits to improve the readability of the manuscript.

1. Please mention the possibility of adv.C in the NaCl samples and mention the impact on the attenuation (possibly on page 12-13). I feel that this is important as the audience of this journal might otherwise get the impression that adv. C. is unique to the organic samples.

   We have added a paragraph on the possible impact of adventitious carbon on the NaCl measurements in Section 3.1.3.

2. I still find the reporting of the binding energy shifts hard to understand for a broader audience. It appears that the relative shifts to 0 mbar are compared in absolute shifts for the different samples. My argument here is, that because the 0mbar spectra are less shifted for the organics as compared to the NaCl - for reasons related to the sample holder and/or the sample - the shift when introducing H2O vapour can not be as large as that observed for NaCl. In this understanding the H2O (adsorbed and in the gas-phase) "only" reduces the shift caused by charging - because it establishes a better charge transfer (conductivity) via the gas phase and via the interface. The BE can not change further than typical values for well conducting surfaces. I think we agree on this, don't we? Looking at Figure 3, I get the impression that the shifts you observe (at low RH) for the organics are already 100% of these expected changes in BE. Increasing the RH further, I would argue that you can not detect further changes in

BE even if more water would adsorb. I don't see this explicitly mentioned or discussed in the paper.

I kindly ask you to mention the reason for the BE shift at 0 mbar more explicitly so that a non-expert audience can follow and then shortly elaborate on the aspect I tried to mention above. I think the reader needs to clearly see how much the 0 mbar BE shifted (due to charging), how much the BE can thus change with increasing RH (going back to the uncharged value) and how much change you observed.

We now explain in Section 2 how binding energies can be shifted relative to literature values due to charging and how this shift can be counteracted with the introduction of a gas phase. For the NaCl measurements, we now note in Section 3.1 that the binding energy we report at 0 mbar is shifted relative to literature values due to charging of the silicon substrate. In Section 3.1.1, we discuss how the introduction of water vapor is expected to reverse some of the charging observed at 0 mbar. For the sucrose and malonic acid measurements, we note that the high conductivity of the gold substrate makes it unlikely that the binding energies will be affected by charging.